# Mutational sources of *trans*-regulatory variation affecting gene expression in *Saccharomyces cerevisiae*

Fabien Duveau[1,2]*, Petra Vande Zande[3], Brian PH Metzger[1], Crisandra J Diaz[3], Elizabeth A Walker[1], Stephen Tryban[1], Mohammad A Siddiq[1], Bing Yang[3], Patricia J Wittkopp[1,3]*

[1]Department of Ecology and Evolutionary Biology, University of Michigan, Ann Arbor, United States; [2]Laboratory of Biology and Modeling of the Cell, Ecole Normale Supérieure de Lyon, CNRS, Université Claude Bernard Lyon, Université de Lyon, Lyon, France; [3]Department of Molecular, Cellular, and Developmental Biology, University of Michigan, Ann Arbor, United States

**Abstract** Heritable variation in a gene's expression arises from mutations impacting *cis*- and *trans*-acting components of its regulatory network. Here, we investigate how *trans*-regulatory mutations are distributed within the genome and within a gene regulatory network by identifying and characterizing 69 mutations with *trans*-regulatory effects on expression of the same focal gene in *Saccharomyces cerevisiae*. Relative to 1766 mutations without effects on expression of this focal gene, we found that these *trans*-regulatory mutations were enriched in coding sequences of transcription factors previously predicted to regulate expression of the focal gene. However, over 90% of the *trans*-regulatory mutations identified mapped to other types of genes involved in diverse biological processes including chromatin state, metabolism, and signal transduction. These data show how genetic changes in diverse types of genes can impact a gene's expression in *trans*, revealing properties of *trans*-regulatory mutations that provide the raw material for *trans*-regulatory variation segregating within natural populations.

**\*For correspondence:**
fabien.duveau@ens-lyon.fr (FD);
wittkopp@umich.edu (PJW)

## Introduction

The regulation of gene expression is a complex process, essential for cellular function, that impacts development, physiology, and evolution. Expression of each gene is regulated by its *cis*-regulatory DNA sequences (e.g. promoters, enhancers) interacting either directly or indirectly with *trans*-acting factors (e.g. transcription factors, signaling pathways) encoded by genes throughout the genome. Genetic variants affecting both *cis*- and *trans*-acting components of regulatory networks contribute to expression differences within and between species (*Albert and Kruglyak, 2015*; *Barbeira et al., 2018*; *Ferraro et al., 2020*; *Gamazon et al., 2018*; *Oliver et al., 2005*). This regulatory variation arises the same way as genetic variation affecting any other quantitative trait: new mutations generate variation in gene expression and selection favors the transmission of some genetic variants over others, giving rise to polymorphism within a species and divergence between species. Because new mutations are the raw material for this polymorphism and divergence, knowing how new mutations impact gene expression is essential for understanding how gene regulation evolves (reviewed in *Hill et al., 2021*). Targeted mutagenesis has been used to systematically examine the effects of individual mutations in *cis*-regulatory sequences for a variety of elements in a variety of species (*Hornung et al., 2012*; *Kwasnieski et al., 2012*; *Maricque et al., 2017*; *Melnikov et al., 2012*; *Metzger et al., 2015*; *Patwardhan et al., 2009*; *Sharon et al., 2012*), but such targeted approaches are not well-suited for surveying the effects of new *trans*-regulatory mutations because *trans*-

regulatory mutations can be located virtually anywhere within the genome. Consequently, we know comparatively little about the genomic sources, molecular mechanisms of action and evolutionary contributions of individual *trans*-regulatory mutations.

Genetic mapping experiments and genome-wide association studies (GWAS) have shown that gene expression is a highly polygenic trait, with hundreds of genetic variants typically associated with natural variation in expression levels of each gene (*Albert et al., 2018*; *Metzger and Wittkopp, 2019*; *Sinnott-Armstrong et al., 2021*). Although these studies often lack the resolution to identify individual genetic changes affecting expression, most of this variation maps far from the gene whose expression it affects and is therefore likely to have *trans*-acting effects. *Trans*-acting variants segregating in natural populations are most often expected to affect transcription factors (*Albert et al., 2018*; *Lewis et al., 2014*), but they can also alter genes encoding signaling proteins, chromatin modifiers, metabolic enzymes, or any other gene product that can influence the availability, accessibility, or activity of transcription factors (*Lutz et al., 2019*; *Mehrabian et al., 2005*; *Schadt et al., 2005*; *Yvert et al., 2003*). Indeed, the recently proposed omnigenic model emphasizes the interconnectedness of regulatory networks controlling transcription to help explain the highly polygenic nature of diverse quantitative traits.

Despite the vast potential target size for *trans*-regulatory mutations (*Hill et al., 2021*), regions of the genome most likely to harbor mutations affecting a particular gene's expression might be predictable from knowledge of its regulatory network. Among eukaryotes, the set of genes and interactions regulating gene expression in *trans* is perhaps best understood in the baker's yeast *Saccharomyces cerevisiae* (*Hughes and de Boer, 2013*): networks of regulatory connections (*Teixeira et al., 2018*) have been inferred from experiments that profile the transcriptional effects of gene deletions (*Hughes et al., 2000*; *Jackson et al., 2020*; *Kemmeren et al., 2014*), map binding sites for transcription factors (*Rhee and Pugh, 2011*; *Zheng et al., 2010*; *Zhu et al., 2009*), identify protein-protein interactions (*Gavin et al., 2002*; *Liu et al., 2020*; *Tarassov et al., 2008*), and test pairs of genes for genetic interactions (*Costanzo et al., 2016*; *van Leeuwen et al., 2016*). However, the extent to which the genomic sources of *trans*-regulatory mutations can be predicted from such networks is generally unknown (*Flint and Ideker, 2019*). In addition, the extent to which the genomic distribution of new mutations predicts the genomic distribution of natural polymorphisms is also unclear because mutations that are strongly deleterious might rarely be found circulating within a population as standing genetic variation. For example, mutations in coding sequences might often impact gene expression but might also tend to be more pleiotropic and thus more deleterious than mutations in non-coding regions of these genes. Comparing the genomic distribution of mutations that have not experienced natural selection to the genomic distribution of polymorphisms that have can reveal such differences between the possible and actual sources of variation in gene expression in the wild.

Systematic studies of new mutations identifying and characterizing the effects of individual genetic changes are thus an important complement to GWAS describing the polygenic variation segregating within a species. Recently, a chemical mutagen was used to induce mutations throughout the genome of *S. cerevisiae,* and hundreds of mutant genotypes were collected that all altered expression of the same gene, providing the biological resources needed to systematically characterize properties of new *trans*-regulatory mutations and to test the predictive power of inferred regulatory networks (*Gruber et al., 2012*; *Metzger et al., 2016*). Here, we use genetic mapping, candidate gene sequencing and functional validation to identify 69 *trans*-regulatory mutations that alter expression of the focal gene from this set of mutants and contrast their properties with a comparable set of 1766 mutations that did not affect expression of the focal gene.

Using this collection of individual *trans*-regulatory mutations, we determined how *trans*-regulatory mutations affecting expression of a single gene were distributed within the genome and within a regulatory network. For example, we asked how frequently *trans*-regulatory mutations were located in coding or non-coding sequences because *trans*-regulatory variants are often predicted to affect coding sequences (*Hill et al., 2021*) but some non-coding variants have been shown to be associated with *trans*-regulatory effects on gene expression (*GTEx Consortium, 2020*; *Yao et al., 2017*; *Yvert et al., 2003*). We also asked whether genes encoding transcription factors were the primary source of *trans*-regulatory variation, which is often assumed (*Albert et al., 2018*; *Lewis et al., 2014*) despite case studies identifying *trans*-regulatory variants in genes encoding proteins with other functions (*Lutz et al., 2019*; *Mehrabian et al., 2005*; *Schadt et al., 2005*; *Yvert et al., 2003*). To

determine how well an inferred regulatory network can predict genomic sources of expression changes, we mapped the *trans*-regulatory mutations to a network of transcription factors predicted by functional genomic data to regulate expression of the focal gene and examined the molecular functions and biological processes impacted by *trans*-regulatory mutations that did not map to genes in this network. By systematically examining the properties and identity of new *trans*-regulatory mutations, this work fills a key gap in our understanding of how expression differences arise and may help predict sources of *trans*-regulatory variation segregating in natural populations. Indeed, we found that the genomic distribution of new *trans*-regulatory mutations overlaps significantly with the genomic distribution of *trans*-regulatory variants segregating among wild isolates of *S. cerevisiae* that affect expression of the same gene (*Metzger and Wittkopp, 2019*), suggesting that the mutational process generating new *trans*-regulatory variation significantly shaped the regulatory variation we see in the wild.

## Results

### Genetic mapping of *trans*-regulatory mutations

To characterize properties of new *trans*-regulatory mutations affecting expression of a focal gene, we took advantage of three previously collected sets of haploid mutants that all showed altered expression of the same reporter gene (*Figure 1A*, *Gruber et al., 2012*; *Metzger et al., 2016*). This reporter gene ($P_{TDH3}$-YFP) encodes a yellow fluorescent protein whose expression is regulated by the *S. cerevisiae TDH3* promoter, which natively drives constitutive expression of a glyceraldehyde-3-phosphate dehydrogenase involved in glycolysis and gluconeogenesis (*McAlister and Holland, 1985*). The mutation rate was increased to obtain these mutants by exposure to the chemical mutagen ethyl methanesulfonate (EMS), which induces primarily G:C to A:T point mutations randomly throughout the genome (*Shiwa et al., 2012*). The dose of EMS used in these studies was chosen so that most mutants with a detectable change in $P_{TDH3}$-YFP expression should have only one mutation causing this change in expression among the mutations they carry (*Metzger et al., 2016*; *Gruber et al., 2012*). Together, these collections contain ~1500 mutants isolated irrespective of their fluorescence levels ('unenriched' mutants) and ~1200 mutants isolated after enriching for cells with the largest changes in fluorescence (*Figure 1A*, see *Figure 1—figure supplement 1* for a diagram showing the number of mutants and mutations included at each step of the study). When we started this work, expression level of $P_{TDH3}$-YFP in these mutant genotypes had been described (*Gruber et al., 2012*; *Metzger et al., 2016*), but the specific mutations present within each mutant as well as which mutation(s) alter(s) $P_{TDH3}$-YFP expression in each genotype were unknown.

From these collections, we selected 82 EMS-treated mutants for genetic mapping to identify individual causal mutations (*Figure 1A*, *Figure 1—figure supplement 1*). Sanger sequencing of the reporter gene in these mutants showed that none had mutations in the *TDH3* promoter or any other part of the reporter gene, indicating that they harbored mutations affecting $P_{TDH3}$-YFP expression in *trans*. Thirty-nine of these mutants were selected based on previously published fluorescence data, with 11 mutants selected from the collections enriched for large effects (red points in *Figure 1B,C*) and 28 mutants selected from the unenriched collection (red points in *Figure 1D*). Each selected mutant showed changes in average YFP fluorescence greater than 1% relative to the un-mutagenized progenitor strain. Another 197 mutants from the unenriched collection (blue points in *Figure 1D*) were subjected to a secondary fluorescence screen, from which an additional 43 mutants with a change in fluorescence greater than 1% (red points in *Figure 1E*) were chosen. Overall, the 82 mutants were selected randomly from the 528 EMS mutants that showed statistically significant fluorescence changes greater than 1% relative to wild-type ($p < 0.05$, see Methods and *Figure 1* legend for a description of the statistical tests). A 1% change in YFP fluorescence has previously been shown to correspond to a ~3% change in YFP mRNA abundance (see *Duveau et al., 2018*), although changes in fluorescence caused by *trans*-regulatory mutations in these mutants could affect either transcription driven by the *TDH3* promoter or post-transcriptional regulation of YFP synthesis or stability.

To identify mutations within the 82 selected EMS mutants, and to determine which of these mutation(s) were most likely to affect YFP expression in each mutant, we performed bulk-segregant analysis followed by whole-genome sequencing (BSA-Seq) as described in *Duveau et al., 2014* with

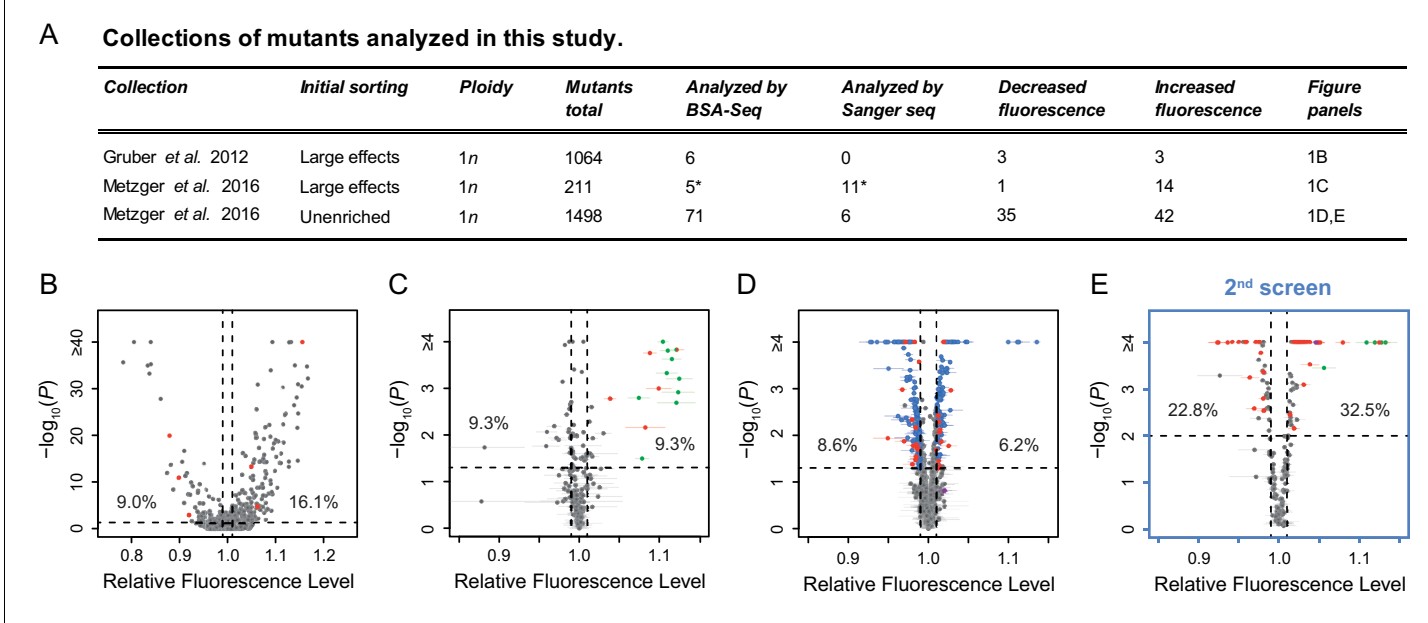

**Figure 1.** Mutant strains analyzed with altered expression of a $P_{TDH3}$-YFP reporter gene. (**A**) Summary of the three previously published collections of *S. cerevisiae* mutants obtained by ethyl methanesulfonate (EMS) mutagenesis of a haploid strain expressing a yellow fluorescent protein (YFP) under control of the *TDH3* promoter. *One mutant is included in both columns because it was analyzed both by BSA-Seq and Sanger sequencing. (**B–D**) Previously published fluorescence levels (x-axis) and statistical significance of the difference in median fluorescence between each mutant and the un-mutagenized progenitor strain (y-axis) are shown for mutants analyzed in (**B**) *Gruber et al., 2012* and (**C,D**) *Metzger et al., 2016*. (**B**) Collection of 1064 mutants from *Gruber et al., 2012* enriched for mutations causing large fluorescence changes. p-values were computed using *Z*-tests in this study, based on one measure of fluorescence for each mutant and 30 measures of fluorescence for the progenitor strain. (**C**) Collection of 211 mutants from *Metzger et al., 2016* enriched for mutations causing large fluorescence changes. (**D**) Collection of 1498 mutants from *Metzger et al., 2016* obtained irrespective of their fluorescence levels (unenriched mutants). (**E**) A new fluorescence dataset for 197 unenriched mutants from *Metzger et al., 2016* (blue in panel D) that were reanalyzed in a 2nd screen as part of this study. (**C–E**) Four replicate populations were analyzed for each mutant. Error bars show 95% confidence intervals of fluorescence levels measured among these replicates. p-values were obtained using the permutation tests described in Methods. (**B–E**) Mutants analyzed by BSA-Seq are highlighted in red. All of these mutants showed fluorescence changes greater than 0.01 (vertical dotted lines) and p-value below 0.05 (horizontal dotted lines); percentages of all mutants that met these selection criteria in each collection are also shown. Mutants selected for Sanger sequencing of the *ADE4*, *ADE5*, and/or *ADE6* candidate genes are highlighted in green. The mutant analyzed with both BSA-seq and Sanger sequencing is both red and green in panel (**C**). Two mutants selected for Sanger sequencing of the *ADE2* gene are highlighted in purple, one in (**D**) and one in (**E**).

The online version of this article includes the following figure supplement(s) for figure 1:

**Figure supplement 1.** Diagram showing the number of mutant strains and mutations considered at each step of the study.

minor modifications (see Methods). Briefly, each mutant strain was crossed to a common mapping strain expressing the $P_{TDH3}$-YFP reporter gene, and large populations of random haploid spores were isolated after inducing meiosis in the resulting diploids (*Figure 2A*). For each of the 82 segregant populations, a low fluorescent bulk and a high fluorescent bulk of ~1.5 x $10^5$ cells each were isolated using fluorescence-activated cell sorting (FACS) (*Figure 2B*). Genomic DNA extracted from each bulk was then sequenced to an average coverage of ~105x (ranging from 75x to 134x among samples, *Supplementary file 1*) to identify the mutations present within each mutant genotype and to quantify the frequency of mutant and non-mutant alleles in both bulks (*Figure 2C*). A mutation causing a change in fluorescence is expected to be found at different frequencies in the two populations of segregant cells. Conversely, a mutation with no effect on fluorescence that is not genetically linked to a mutation affecting fluorescence is expected to be found at similar frequencies in these two populations.

Using a stringent approach for calling sequence variants (see Methods), we identified a total of 1819 mutations in the BSA-Seq data from the 76 mutants from *Metzger et al., 2016*; *Supplementary file 2*, *Figure 1—figure supplement 1*, among which 1768 mutations (97.2%) were single nucleotide changes (*Figure 2D*). Of these single nucleotide changes, 96.3% were one of the

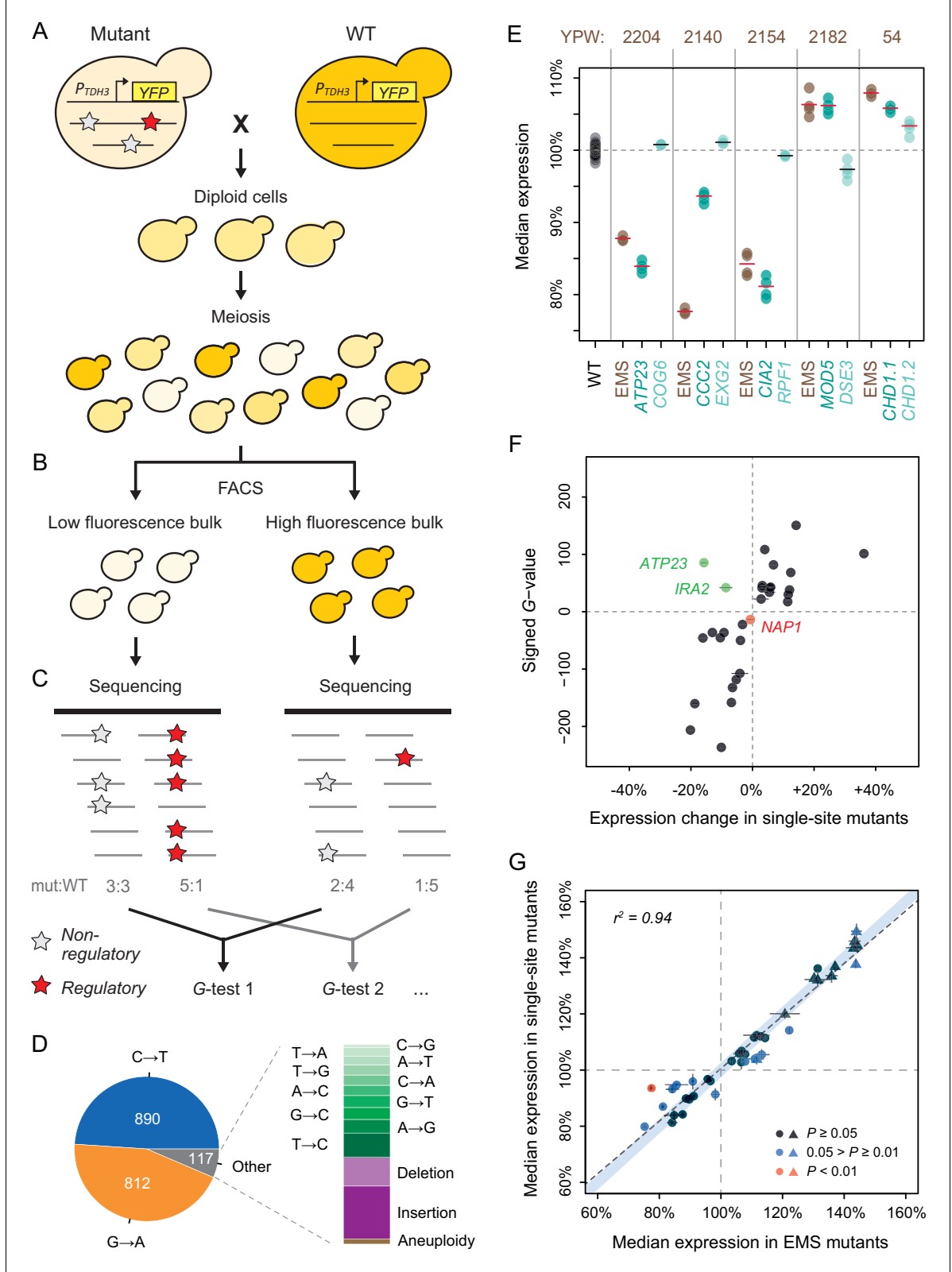

**Figure 2.** Genetic mapping and functional testing of *trans*-regulatory mutations affecting $P_{TDH3}$-*YFP* expression. (A–C) Overview of the BSA-Seq approach. (A) Crossing scheme used to map mutations in each EMS mutant strain by crossing to an un-mutagenized strain expressing $P_{TDH3}$-*YFP*. Stars indicate hypothetical mutations. (B) Isolation of two bulks of haploid segregants with high and low fluorescence levels (see Methods). (C) Estimation of allele frequencies in each bulk using high-throughput sequencing. A mutation without effect on fluorescence is found at similar frequencies in the two

*Figure 2 continued on next page*

*Figure 2 continued*

bulks (white stars). A mutation affecting fluorescence or genetically linked to a mutation affecting fluorescence is found at different frequencies between the two bulks (red stars). (D) Type of mutations identified in BSA-Seq data for the 76 mutants from *Metzger et al., 2016*. (E) Median expression of $P_{TDH3}$-YFP is shown for the wild-type (WT) progenitor strain (black), for five EMS mutants (brown) with two linked mutations associated with fluorescence in BSA-Seq data and for 10 single-site mutants (turquoise) carrying one of the two linked mutations in the five EMS mutants. Single-site mutants are grouped in pairs next to the EMS mutant carrying the same mutations and are named after the gene that they affect. Expression levels are expressed relative to the wild-type progenitor strain. For each strain, dots represent the median expression measured for each replicate population and tick marks represent the mean of median expression from replicate populations. (F) Effects of mutations associated with fluorescence in BSA-Seq experiments tested in single-site mutants. X-axis: Effect of each mutation on expression measured in a single site mutant and relative to the wild-type progenitor strain. Error bars are 95% confidence intervals obtained from at least four replicate populations. Y-axis: G statistics of the tests used to compare the frequencies of each mutation between the two bulks in BSA-Seq experiments, with a negative sign if the mutation was more frequent in the low fluorescence bulk and a positive sign if the mutation was more frequent in the high fluorescence bulk. One single-site mutant (*NAP1*, red) showed no significant change in expression relative to the wild-type progenitor strain (*t*-test, p-value > 0.05); the mutation it carries is therefore considered to be a false positive in the BSA-seq data. For two other single-site mutants (*ATP23* and *IRA2*, green), the expression changes were not in the same direction as predicted by the signed G-values. (G) $P_{TDH3}$-YFP expression levels in single-site mutants and in EMS mutants sharing the same mutation. Data points represent median expression levels of 40 EMS mutants (x-axis) and 40 single-site mutants (y-axis) measured by flow cytometry in four replicate populations. Circles: mutations identified by BSA-Seq. Triangles: mutations identified by sequencing candidate genes. Error bars: 95% confidence intervals of expression levels obtained from replicate populations. Data points are colored based on the p-values of permutation tests used to assess the statistical significance of expression differences between each single site mutant and the EMS mutant carrying the same mutation (see *Figure 2—figure supplement 5* for details). The light blue area represents the 95% confidence interval of expression differences between genetically identical samples across the whole range of median expression values. This confidence interval was calculated from a null distribution described in *Figure 2—figure supplement 5A*. (E–G) Expression levels are expressed on a scale linearly related to *YFP* mRNA levels and relative to the median expression of the wild-type progenitor strain (see Materials and methods).

The online version of this article includes the following figure supplement(s) for figure 2:

**Figure supplement 1.** Number of mutations per strain identified from BSA-Seq data.

**Figure supplement 2.** Magnitude of expression changes in EMS mutants depending on the number of mutations associated with fluorescence in BSA-Seq experiments.

**Figure supplement 3.** Relationship between the number of mutations per EMS mutant strain and the absolute expression change relative to the progenitor strain.

**Figure supplement 4.** Effects of individual mutations in purine biosynthesis genes on YFP expression levels differ among promoters.

**Figure supplement 5.** Factors contributing to expression differences observed between EMS and single-site mutants.

two types of point mutations (G:C to A:T transitions) known to be primarily induced by EMS (*Shiwa et al., 2012*). Forty-eight small indels and three aneuploidies, which could have arisen spontaneously or been introduced by EMS, were also identified. Of these three mutants with aneuploidies, two were found to have an extra copy of chromosome I and one was found to have an extra copy of chromosome V based on ~1.5-fold higher sequencing coverage of these chromosomes relative to the rest of the genome in the BSA-seq data from segregant populations (shown in *Supplementary file 3*). We identified an average of 23.9 mutations per strain, which is within the 95% confidence interval of 21–45 mutations per strain estimated previously from the frequency of canavanine resistant mutants (*Metzger et al., 2016*). Surprisingly, the number of mutations per strain did not follow a Poisson distribution: we observed more strains with a number of mutations far from the average than expected for a Poisson process (p-value < $10^{-5}$, resampling test; *Figure 2—figure supplement 1*), which could be explained by cell-to-cell heterogeneity in DNA repair after exposure to the mutagen (*Liu et al., 2019*; *Uphoff et al., 2016*).

At least one mutation was significantly associated with fluorescence in 46 of the mutants analyzed based on likelihood ratio tests (G-tests described in Materials and methods, *Supplementary file 2*), with a total of 67 mutations associated with fluorescence identified among these mutants (*Figure 1—figure supplement 1*), including all three aneuploidies (*Supplementary file 3*). Twenty-nine mutants had a single mutation associated with fluorescence, 13 mutants had two associated mutations, and 4 mutants had three associated mutations. However, 8 of the 13 mutants with two associated mutations and all four mutants with three associated mutations showed linkage (genetic distance below 25 cM) between at least two of the mutations associated with fluorescence (*Supplementary file 4*, *Figure 1—figure supplement 1*), suggesting that only one of the linked mutations might impact fluorescence in each of these mutants. To determine whether one linked mutation was more likely to impact fluorescence than the others, we compared the magnitude of allele-frequency difference between the high and low fluorescence pools (estimated by the G-value) for each mutation. For 9 of

the 12 mutants with linked mutations, we found that the mutation with the highest $G$-value was significantly more strongly associated with fluorescence than the linked mutation(s) (resampling test: p < 0.05, *Supplementary file 4*), suggesting that this mutation was responsible for the fluorescence change. For the other three mutants, none of the linked mutations showed stronger evidence of impacting fluorescence than the others (resampling test: p > 0.05, *Supplementary file 4*).

The remaining 36 mutants did not have any mutations significantly associated with fluorescence (*Supplementary file 2*, *Figure 1—figure supplement 1*). These mutants tended to show smaller changes in fluorescence than mutants with one or more associated mutations (*Figure 2—figure supplement 2*), suggesting that our power to map mutations causing 1% changes in fluorescence might have been lower than anticipated. These 36 mutants might also harbor multiple mutations with small effects on expression, each of which was below our detection threshold. Consistent with this possibility, we observed a small but significant correlation ($r^2$ = 0.127, p = 0.03) between the total number of mutations in these 36 EMS mutants and their expression level (*Figure 2—figure supplement 3*). It is also possible that we failed to find associated mutations in some of these mutants because their change in fluorescence was initially overestimated by the 'winner's curse' (*Xiao and Boehnke, 2009*). Accordingly, 71% of mutants selected for mapping after two independent fluorescence screens had at least one mutation significantly associated with fluorescence compared to only 30% of mutants selected after a single fluorescence screen. Some changes in fluorescence observed in these 36 mutants might also have been caused by non-genetic variation and/or undetected mutations.

## Additional *trans*-regulatory mutations identified by sequencing candidate genes

We noticed in the BSA-seq data that three mutations increasing fluorescence more than 5% relative to the un-mutagenized progenitor strain mapped to two genes (*ADE4* and *ADE5*) in the same biochemical pathway (de novo purine biosynthesis) (*Supplementary file 2*). We therefore used Sanger sequencing to test whether these genes or other genes in this pathway were also mutated in 15 additional EMS mutants with fluorescence at least 5% higher than the progenitor strain. We first looked for mutations in *ADE4*, then *ADE5* if no mutation was found in *ADE4*, and then *ADE6* if no mutation was found in the other genes. At least one nonsynonymous mutation was identified by Sanger sequencing in one of these three genes in 14 of the 15 EMS mutants (green points in *Figure 1C,E*; *Supplementary file 5*, *Figure 1—figure supplement 1*). For the remaining mutant (brown point in *Figure 1E*), we sequenced a fourth purine biosynthesis gene, *ADE8*, but again found no mutation. In two additional EMS mutants with smaller increases in fluorescence (2.1% and 4.6%, purple points in *Figure 1D,E*) and a reddish color characteristic of *ADE2* loss of function mutants (*Roman, 1956*), we found nonsynonymous mutations in *ADE2* by Sanger sequencing (*Supplementary file 5*, *Figure 1—figure supplement 1*). Follow-up experiments showed that mutations in *ADE2*, *ADE5*, and *ADE6* did not increase YFP fluorescence driven by two other promoters ($P_{RNR1}$ and $P_{STM1}$), suggesting that mutations in the purine biosynthesis pathway affected expression of $P_{TDH3}$-YFP through mechanisms mediated by the *TDH3* promoter rather than YFP (*Figure 2—figure supplement 4*). Taken together, these data suggest that genes in the purine biosynthesis pathway are the predominant mutational source of large increases in *TDH3* expression.

## Functional testing confirms effects of *trans*-regulatory mutations identified by genetic mapping and candidate gene sequencing

To determine whether mutations statistically associated with fluorescence in the BSA-seq data actually affected expression of $P_{TDH3}$-YFP, we introduced 34 of the 67 associated mutations individually into the fluorescent progenitor strain using scarless genetic engineering approaches (*Supplementary file 6*, *Figure 1—figure supplement 1*). We also used scarless genome editing to create single-site mutants for 11 of the 17 additional mutations identified in purine biosynthesis genes by Sanger sequencing (*Supplementary file 5*, *Supplementary file 6*, *Figure 1—figure supplement 1*). Fluorescence of these engineered strains (called 'single-site mutants' hereafter) was then quantified by flow cytometry in parallel with fluorescence of the EMS mutant carrying the same associated mutation as well as the un-mutagenized progenitor strain, with four replicate populations

analyzed for each genotype. Fluorescence values were then transformed into estimates of YFP abundance as described in the Methods.

Of the 24 mutations without linked variants in EMS mutants that were tested in single-site mutants, 23 (96%) caused a significant change in expression (p < 0.05, permutation test, *Supplementary file 6*), suggesting a ~4% false positive rate in our BSA-Seq experiment. In addition, all 11 single-site mutants with mutations in purine biosynthesis genes identified by Sanger sequencing showed statistically significant effects on fluorescence relative to the un-mutagenized progenitor strain (all increased fluorescence, p < 0.05, permutation test, *Supplementary file 6*). The remaining 10 mutations tested in single-site mutants were from five of the EMS mutants with two linked mutations associated with fluorescence. Each of these mutations was introduced separately into a single-site mutant to independently measure its effect on expression. For four of these five pairs of linked mutations, only one of the two single-site mutants showed a significant change in expression relative to the progenitor strain (*Figure 2E*). In each case, the single-site mutant and the EMS mutant showed changes in expression in the same direction relative to the progenitor strain (*Figure 2E*). The mutation affecting expression was always the mutation with the larger *G*-value in the BSA-Seq data, consistent with the results of the statistical tests described above (*Supplementary file 4*). In the last case (YPW54 in *Figure 2E*), both mutations affected expression in the single-site mutants, consistent with our inability to statistically predict which mutation was more likely to impact expression from the BSA-Seq data for this mutant as well as both mutations being nonsynonymous changes in the same gene (*CHD1*) (*Supplementary file 4*). The BSA-seq data also accurately predicted whether a mutation increased or decreased fluorescence for 27 (93%) of the 29 mutations with significant effects on fluorescence in single-site mutants (*Figure 2F*). For the other two mutations, effects on expression in the same direction were observed in the single-site mutants and the corresponding EMS mutants (*Supplementary file 5*), suggesting that the different growth conditions used for the mapping experiment (see Methods) might have modified the effects of these mutations.

Comparing $P_{TDH3}$-YFP expression in the 40 single-site mutants that significantly altered fluorescence to that in the 40 EMS mutants from which these mutations were identified showed that expression was very similar overall between single-site and EMS mutants sharing the same mutation (*Figure 2G*, linear regression: $r^2$ = 0.944, p = 2.4 x $10^{-25}$), although significant differences in expression were observed for some pairs (*Figure 2G*, *Figure 2—figure supplement 5*). The linear correlation between the expression of single-site mutants and EMS mutants remained strong when mutations identified by sequencing candidate genes (triangles in *Figure 2G*) were excluded ($r^2$ = 0.854, p = 5.5 x $10^{-13}$). These data suggest that (1) the vast majority of the mutations we identified by genetic mapping and candidate gene sequencing do indeed have *trans*-regulatory effects on expression of $P_{TDH3}$-YFP and (2) the majority of EMS mutants analyzed had a single mutation that was primarily, if not solely, responsible for the observed change in $P_{TDH3}$-YFP expression.

## Properties of *trans*-regulatory mutations affecting expression driven by the *TDH3* promoter

In all, 69 mutations showed evidence of affecting $P_{TDH3}$-YFP expression in *trans* (*Figure 1—figure supplement 1*), including 3 aneuploidies and 66 point mutations. Fifty-two of these mutations were identified by genetic mapping (*Supplementary file 7*) and 17 were identified by sequencing candidate genes (*Supplementary file 5*). Twelve of the mutations identified by genetic mapping were genetically linked to one or more other mutations but showed stronger evidence of affecting $P_{TDH3}$-YFP expression than the linked mutation(s) in statistical and/or functional tests described above (*Supplementary file 3*). To identify trends in the properties of these 69 *trans*-regulatory mutations, we compared them to 1766 mutations considered non-regulatory regarding $P_{TDH3}$-YFP expression because they showed no significant association with expression of the reporter gene in the BSA-Seq experiment (*G*-test: p > 0.01, *Figure 1—figure supplement 1*). To be conservative, eight mutations that showed a marginally significant association with expression (*G*-test: 0.001 < p < 0.01) as well as 15 mutations associated with expression only because of genetic linkage were excluded from further analyses.

First, we asked whether the mutational spectra of *trans*-regulatory mutations differed from non-regulatory mutations (*Figure 3A*). We found that G:C to A:T transitions most commonly introduced by EMS occurred at similar frequencies in the two groups (*G*-test, p = 0.84). No indels were associated with expression in the BSA-seq data (*Supplementary file 7*), which was not statistically different

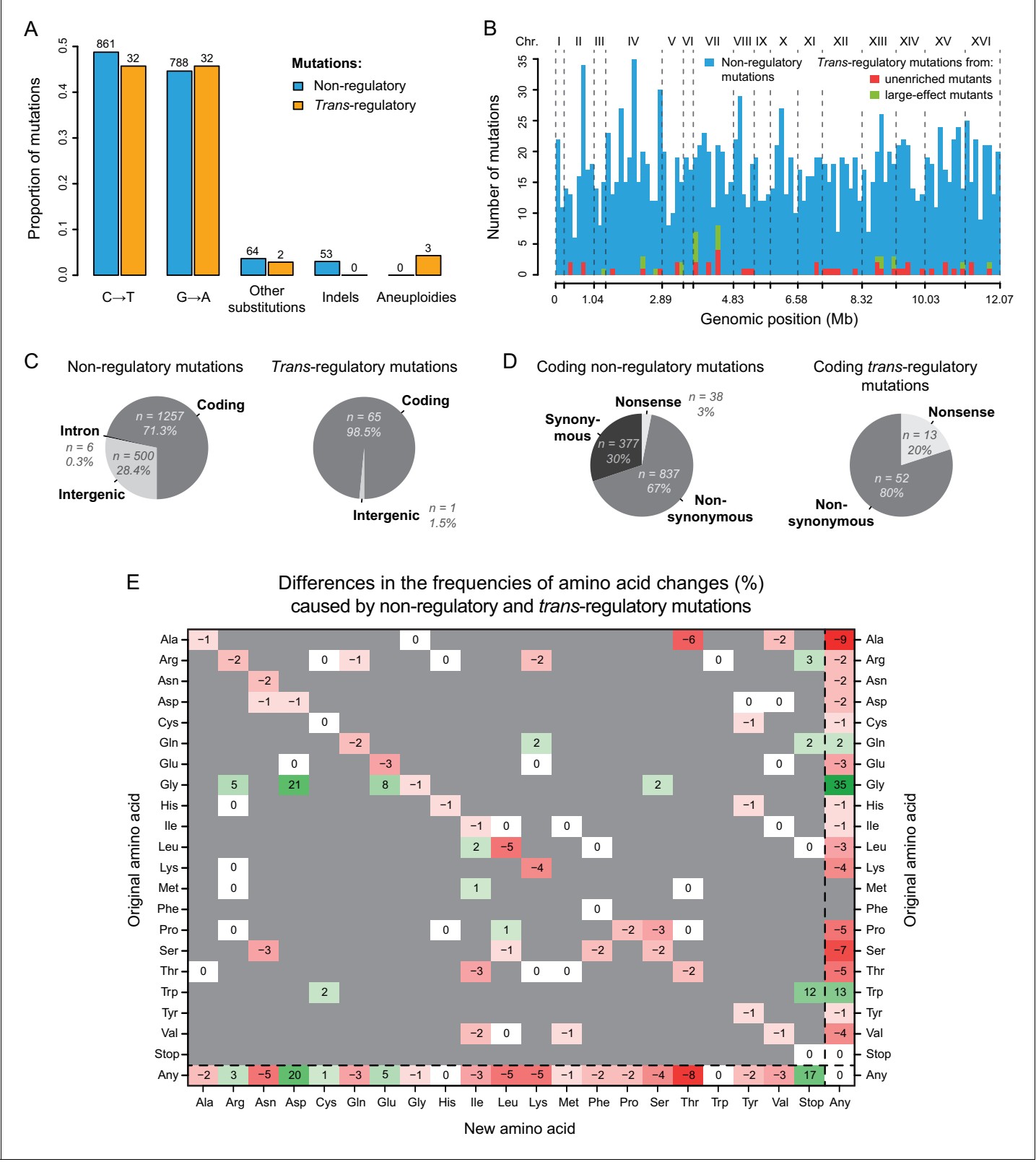

**Figure 3.** Contrasting properties of *trans*-regulatory and non-regulatory mutations. (**A**) Proportions of different types of mutations in a set of 1766 non-regulatory mutations (blue) and in a set of 69 *trans*-regulatory mutations (orange). Numbers of mutations are indicated above bars. (**B**) Distributions of non-regulatory and *trans*-regulatory point mutations along the yeast genome. A total of 1766 non-regulatory mutations are shown in blue, 44 *trans*-regulatory mutations that were identified from the collections of unenriched mutants in *Metzger et al., 2016* are shown in red and 22 *trans*-regulatory

*Figure 3 continued on next page*

*Figure 3 continued*

mutations that were identified from the collections of mutants enriched for large expression changes in *Gruber et al., 2012* and in *Metzger et al., 2016* are shown in green. (C) Proportions of non-regulatory (left) and *trans*-regulatory (right) mutations affecting either coding sequences, introns or intergenic regions. (D) Proportions of coding non-regulatory (left) and coding *trans*-regulatory (right) mutations that either introduce an early stop codon (nonsense), that substitute one amino acid for another (nonsynonymous) or that do not change the amino acid sequence (synonymous). (E) Frequency of all amino acid changes induced by *trans*-regulatory mutations as compared to non-regulatory mutations. Each entry of the table represents the difference of frequency (percentage) between non-regulatory and *trans*-regulatory mutations that are changing the amino acid shown on the y-axis into the amino acid shown on the x-axis. For instance, the −6 on the first row indicates that the proportion of mutations changing an Alanine into a Threonine is 6% lower among *trans*-regulatory mutations than among non-regulatory mutations. Shades of red: amino acid changes underrepresented in the set of *trans*-regulatory mutations. Shades of green: amino acid changes overrepresented in the set of *trans*-regulatory mutations. White: amino acid changes equally represented in the *trans*-regulatory and non-regulatory sets of mutations. Gray: amino acid changes not observed in the sets of *trans*-regulatory and non-regulatory mutations. (B–E) The three aneuploidies were excluded for these plots. (D,E) Non-coding mutations were excluded for these plots.

The online version of this article includes the following figure supplement(s) for figure 3:

**Figure supplement 1.** Contrasting properties of non-regulatory and *trans*-regulatory mutations identified by BSA-Seq and of *trans*-regulatory mutations identified by Sanger sequencing of candidate genes.
**Figure supplement 2.** Distributions of *trans*-regulatory and non-regulatory mutations among chromosomes.
**Figure supplement 3.** Statistical significance of the enrichment and depletion of amino acid changes induced by *trans*-regulatory mutations.
**Figure supplement 4.** Statistical significance of the enrichment and depletion of amino acid changes induced by *trans*-regulatory mutations identified by BSA-Seq.

from the frequency of indels among non-regulatory mutations (0% vs 2.7%, *G*-test, p = 0.056). By contrast, aneuploidies were highly over-represented in the set of *trans*-regulatory mutations since all three extra copies of a chromosome observed in the BSA-Seq data were found to be associated with fluorescence (*G*-test, p = 8.6 x 10$^{-6}$); a similar overrepresentation was observed when considering only mutations identified by BSA-Seq (*Figure 3—figure supplement 1A*; *G*-test, p = 3.5 x 10$^{-6}$). We also found a significant difference in the genomic distribution of the two sets of mutations (*G*-test, p = 2.4 x 10$^{-3}$), with non-regulatory mutations appearing to be randomly distributed throughout the genome but *trans*-regulatory mutations enriched on chromosomes VII and XIII (*Figure 3B*, *Figure 3—figure supplement 2*). However, these two chromosomes contain the purine biosynthesis genes in which multiple *trans*-regulatory mutations were identified, and there was no significant difference in genomic distributions between *trans*-regulatory and non-regulatory mutations when mutations in purine biosynthesis genes were excluded (*G*-test, p = 0.35) or when mutations identified by direct sequencing of candidate genes were excluded (*Figure 3—figure supplement 1B*; *G*-test, p = 0.22).

*Trans*-regulatory mutations are often assumed to be located in coding sequences, but they can also be located in non-coding, presumably *cis*-regulatory, sequences of *trans*-acting genes (*Hill et al., 2021*). We therefore asked whether *trans*-regulatory mutations affecting P$_{TDH3}$-YFP expression were more often found in coding or non-coding regions of the genome than expected by chance. Of the 1766 non-regulatory mutations, 1257 (71.3%) were coding mutations located in exons, and 506 (28.7%) were non-coding mutations located in intergenic (n = 500) or intronic (n = 6) regions (*Figure 3C*). This paucity of mutations in introns is consistent with the rarity of introns in *S. cerevisiae*, and the overall frequency of non-coding mutations (28.7%) is similar to the fraction of the *S. cerevisiae* genome (30.6% of 12.1 Mb) considered non-coding (https://www.yeastgenome.org/). By contrast, of the 66 *trans*-regulatory point mutations, only one was located in a non-coding sequence (*Figure 3C*). This non-coding mutation was located in the intergenic sequence between *IOC2* and *KIN2*, presumably affecting expression of one or both genes with a downstream effect on P$_{TDH3}$-YFP expression. The three aneuploidies were excluded from this and subsequent analyses because they affected both coding and non-coding sequences of a large number of genes. The underrepresentation of non-coding changes among regulatory mutations was statistically significant (1.5% of *trans*-regulatory mutations are non-coding *vs* 28.4% of non-regulatory mutations; *G*-test, p = 4.3 x 10$^{-9}$), even when excluding mutations identified by sequencing candidate genes (*Figure 3—figure supplement 1C*; *G*-test, p = 9.1 x 10$^{-7}$). These observations suggest that new *trans*-regulatory mutations affecting P$_{TDH3}$-YFP expression by more than 3% (i.e. fluorescence changes greater than 1%) are more likely to alter coding than non-coding sequences. This enrichment in coding

sequences might be because coding sequences tend to have a higher density of functional sites than non-coding sequences.

Finally, we examined how *trans*-regulatory mutations located in coding sequences impacted the amino acid sequences of the corresponding proteins. Among mutations identified in coding sequences, 100% of the 65 *trans*-regulatory mutations changed the amino acid sequence of proteins compared to only 70% of 1257 non-regulatory mutations (*Figure 3D,G* -test, p = 1.4 x 10$^{-4}$). Limiting this analysis to the 48 *trans*-regulatory mutations identified by BSA-seq also showed an enrichment of mutations changing the amino acid sequence of proteins (*Figure 3—figure supplement 1D,G* -test, p = 5.6 x 10$^{-6}$). This difference was primarily driven by mutations that introduced stop codons (nonsense mutations) rather than mutations that substituted one amino acid for another (nonsynonymous mutations): 20% of *trans*-regulatory mutations in coding sequences were nonsense mutations *versus* 3% of non-regulatory mutations (*Figure 3D*; *G*-test, p = 4.8 x 10$^{-6}$), and 80% of *trans*-regulatory mutations were nonsynonymous versus 67% of non-regulatory mutations (*Figure 3D*; *G*-test, p = 0.07). A similar pattern was observed when considering only *trans*-regulatory mutations identified by BSA-Seq (*Figure 3—figure supplement 1D*). Nonsense mutations always altered an arginine, glutamine, or tryptophan codon (*Figure 3E*), consistent with the structure of the genetic code and the types of mutations induced by EMS (Figure 3—figure supplement 2 in *Metzger et al., 2016*). For nonsynonymous mutations, two types of amino acid changes were particularly enriched among *trans*-regulatory mutations (*Figure 3E*; *Figure 3—figure supplement 3*): 26.2% of *trans*-regulatory mutations changed glycine to aspartic acid *versus* 5.2% of non-regulatory mutations (permutation test, p < 10$^{-4}$), and 10.8% of *trans*-regulatory mutations changed glycine to glutamic acid versus 2.7% of non-regulatory mutations (permutation test, p = 0.0042). As a consequence, mutations altering glycine codons were strongly over-represented in general among *trans*-regulatory mutations (49.2% of *trans*-regulatory mutations *vs* 14.5% of non-regulatory mutations in coding sequences; permutation test, p < 10$^{-4}$). This over-representation remained significant after excluding mutations identified by Sanger sequencing (*Figure 3—figure supplement 1E*, *Figure 3—figure supplement 4*; 41.7% of *trans*-regulatory mutations altering glycine vs 14.5% of non-regulatory mutations, p = 10$^{-4}$). This pattern may be observed because glycine is the smallest amino acid, making its substitution likely to modify protein structure (*Bhate et al., 2002*; *Miller, 2007*). Indeed, glycine is one of the three amino acids with the lowest experimental exchangeability (*Yampolsky and Stoltzfus, 2005*) and mutations affecting glycine codons are enriched among mutations causing human diseases (*Khan and Vihinen, 2007*; *Molnár et al., 2016*; *Vitkup et al., 2003*).

## Regulatory mutations are enriched in a predicted *TDH3* regulatory network

Because of the key role transcription factors play in the regulation of gene expression, and because transcription factors have been shown to be a source of *trans*-regulatory variation in natural populations (*Albert et al., 2018*; *Lewis et al., 2014*), we asked whether *trans*-regulatory mutations affecting P$_{TDH3}$-YFP expression were enriched in genes encoding transcription factors. We found that 5 (7.7%) of the 65 *trans*-regulatory coding mutations mapped to the coding sequence of one of the 212 genes predicted to encode a transcription factor in the YEASTRACT database (*Teixeira et al., 2018*), but this was not significantly more than the 5.6% of non-regulatory coding mutations mapping to these genes (*G*-test: p = 0.52). *Trans*-regulatory coding mutations were also not significantly enriched in transcription factor genes when we excluded the 17 mutations identified by Sanger sequencing (*G*-test: p = 0.22). Not all transcription factors are expected to regulate expression of *TDH3*, however, so we also tested for enrichment of *trans*-regulatory mutations among transcription factors specifically predicted to regulate *TDH3*.

Using information consolidated in the YEASTRACT database (*Teixeira et al., 2018*) that supports evidence of a transcription factor binding to a gene's promoter and regulating its expression, we constructed a network (*Figure 4*) of potential direct regulators of *TDH3* as well as potential direct regulators of these direct regulators (1st and 2nd level regulators of *TDH3*) and asked how often the *trans*-regulatory mutations we identified mapped to these genes. We found that four *trans*-regulatory mutations mapped to three genes in this network, with two mutations affecting the 1st level regulator *TYE7*, one mutation affecting the 1st level regulator *GCR2*, and one mutation affecting the 2nd level regulator *TUP1* (*Supplementary file 7*). This number of mutations mapping to genes in the predicted *TDH3* regulatory network was 12-fold greater than expected by chance (6.1% for *trans*-

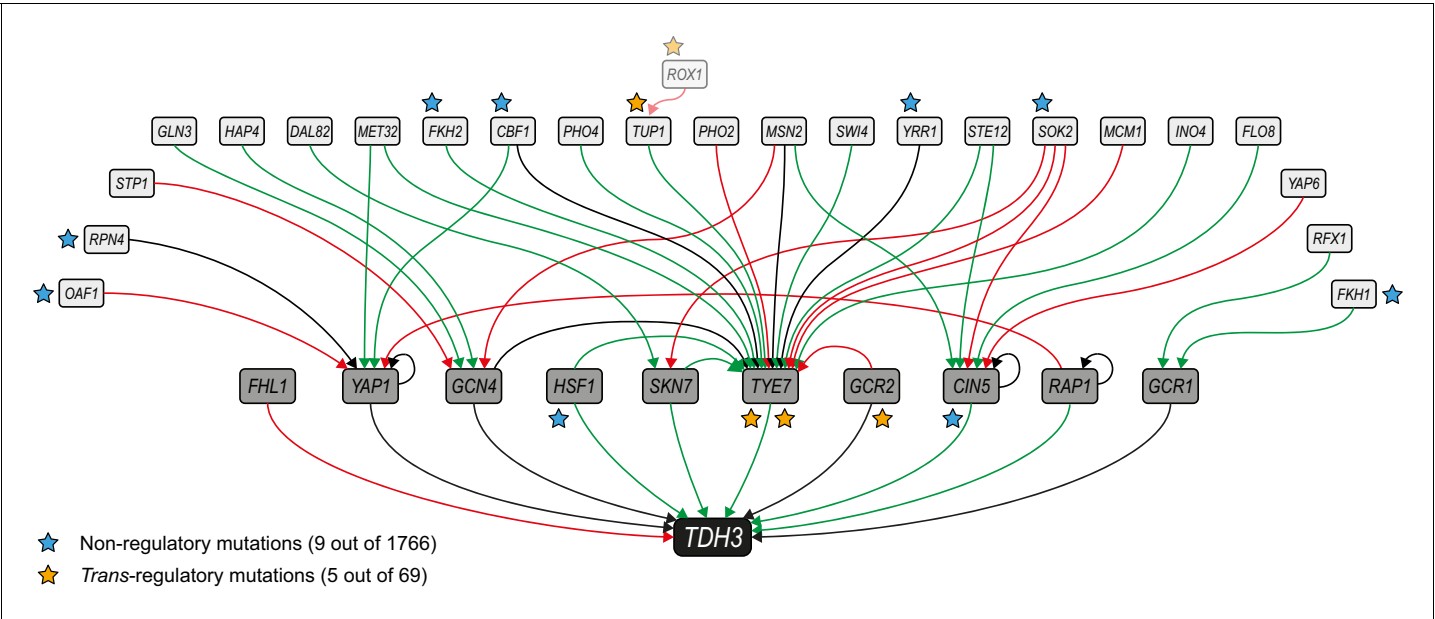

**Figure 4.** Mutations mapping to a predicted *TDH3* regulatory network. The network of inferred interactions between *TDH3* and transcription factors regulating its expression was established using the YEASTRACT repository (*Teixeira et al., 2018*). First level regulators (dark gray boxes) are transcription factors with evidence of binding to the *TDH3* promoter and regulating its expression. Second level regulators (light gray boxes) are transcription factors with evidence of binding to the promoter of at least one first level regulator and regulating its expression. Green arrows: evidence for activation of expression. Red arrows: evidence for inhibition of expression. Black arrows: unknown direction of regulation. Non-regulatory and *trans*-regulatory mutations identified in the network are represented by blue and orange stars, respectively, near the affected genes. *ROX1*, inferred to be a third level regulator, is also shown because a *trans*-regulatory mutation was identified in its coding sequence.

regulatory *vs* 0.5% for non-regulatory mutations; *G*-test, p = 0.0037), or 16-fold greater than expected by chance when excluding mutations identified by Sanger sequencing (8.2% for *trans*-regulatory *vs* 0.5% for non-regulatory mutations; *G*-test, p = 0.0024). Therefore, the inferred regulatory network had predictive power as expected, but the vast majority of *trans*-regulatory coding mutations (61 of 65, or 94%) mapped to genes outside of this network. Only one of these other *trans*-regulatory mutations mapped to a transcription factor. This mutation was a nonsynonymous substitution affecting *ROX1*, which is predicted in the YEASTRACT database to directly regulate expression of the indirect *TDH3* regulator *TUP1*. In other words, *ROX1* is predicted by existing functional genomic data to be a 3rd level regulator of *TDH3* (*Figure 4*). With no other transcription factors harboring a *trans*-regulatory mutation in our dataset, this result suggests that mutations in transcription factors located more than three levels away from *TDH3* in its transcriptional regulatory network are unlikely to be sources of new expression changes driven by the *TDH3* promoter.

## Deleterious effects of mutations in two direct regulators of *TDH3*

Transcription factors encoded by the *TYE7* and *GCR2* genes found to harbor *trans*-regulatory mutations affecting expression of P_{TDH3}-YFP are known to regulate the expression of glycolytic genes (including *TDH3*) by forming a complex with transcription factors encoded by the *RAP1* and *GCR1* genes (*Shively et al., 2019*). Rap1p (*Yagi et al., 1994*) and Gcr1p (*Huie et al., 1992*) are both known to bind directly to the *TDH3* promoter (*Figure 5A*), and mutations in these binding sites cause large decreases in *TDH3* expression (*Metzger et al., 2015*). These observations strongly suggest that mutations in *RAP1* and *GCR1* should also cause detectable changes in *TDH3* expression, yet no mutations were observed in these genes in our set of *trans*-regulatory mutations. To investigate why we did not recover *trans*-regulatory mutations in *RAP1* or *GCR1*, we used error-prone PCR to generate mutant alleles of these genes with mutations in either the promoter or coding sequence of *RAP1* or the second exon of *GCR1*, which includes 99.7% of the *GCR1* coding sequence (*Figure 5B*). Hundreds of these *RAP1* and *GCR1* mutant alleles were then introduced individually into the un-mutagenized strain carrying the P_{TDH3}-YFP reporter gene using CRISPR/Cas9-guided

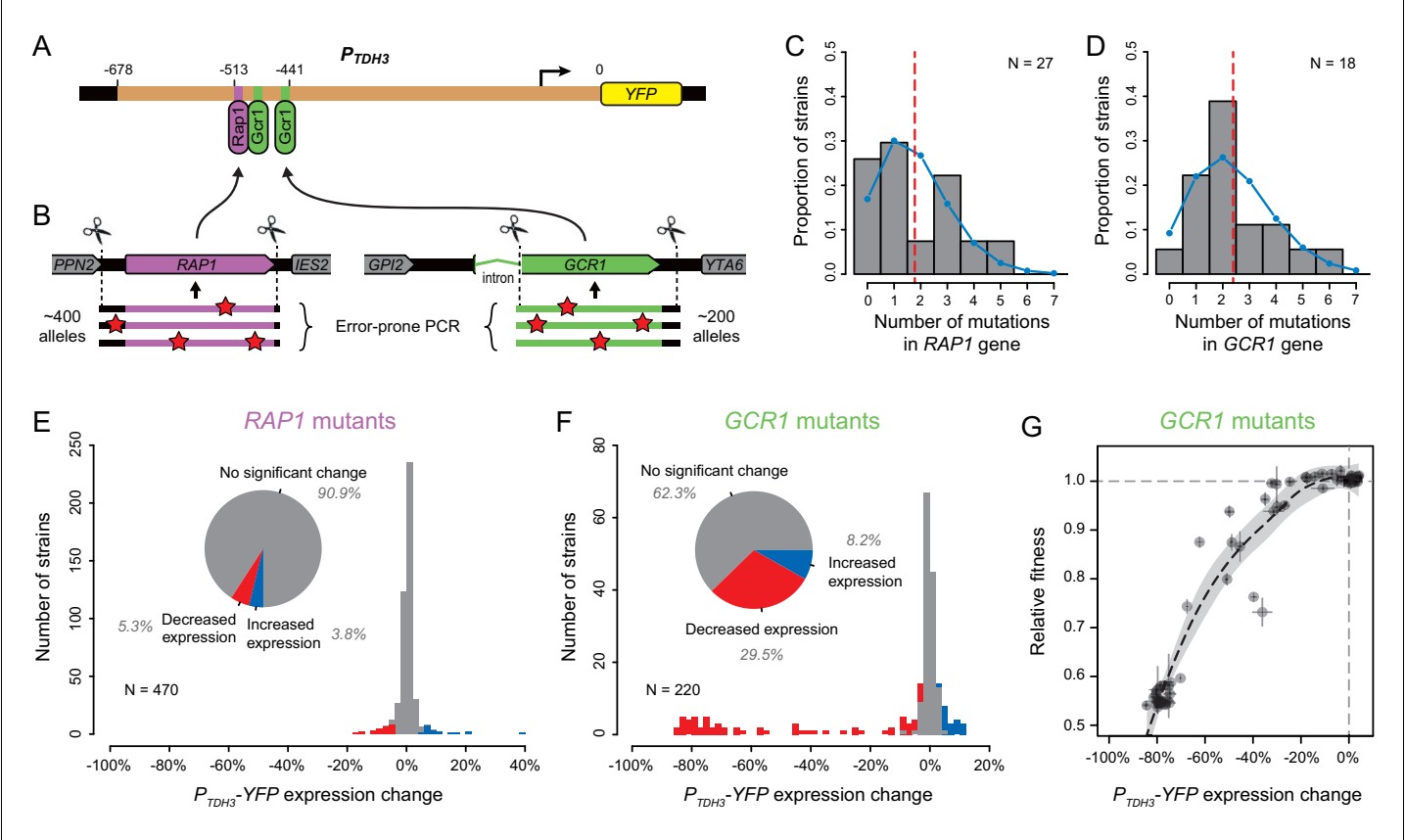

**Figure 5.** Impact of mutations in two direct regulators of the *TDH3* promoter. (A) Schematics of the $P_{TDH3}$-*YFP* reporter gene with locations of three known binding sites for transcription factors Rap1p (purple) and Gcr1p (green) shown in the *TDH3* promoter. (B) Regions of *RAP1* (purple) and *GCR1* (green) genes that were subjected to random mutagenesis using error-prone PCR. 470 *RAP1* mutants and 220 *GCR1* mutants were obtained by integration of random PCR fragments at the native *RAP1* or *GCR1* loci using CRISPR/Cas9 allelic replacement. (C–D) Distributions of the number of mutations per strain identified by Sanger sequencing the mutated regions of (C) *RAP1* in 27 strains or (D) *GCR1* in 18 strains. These data are shown in histograms. Blue curves: Poisson distribution with the same mean as observed in data. Red dotted line: Mean number of mutations among sequenced strains. (E–F) Distributions of $P_{TDH3}$-*YFP* expression changes relative to the un-mutagenized reporter strain measured in four replicate samples for (E) the 470 *RAP1* mutants or (F) the 220 *GCR1* mutants. Fluorescence measures were transformed to be linearly related with *YFP* mRNA levels (see Methods). Red bars: Mutants with significant decrease in median expression greater than 3% relative to the un-mutagenized strain (permutation test, p < 0.05). Blue bars: Mutants with significant increase in median expression greater than 3% relative to the un-mutagenized strain (permutation test, p < 0.05). Pie charts: Proportions of mutants with significant increase in expression (blue), significant decrease in expression (red) and no significant change in expression (gray) relative to the un-mutagenized strain. (G) Relationship between changes in $P_{TDH3}$-*YFP* expression levels (x-axis) and fitness (y-axis) measured in 62 *GCR1* mutants. Expression changes and fitness are both expressed relative to the un-mutagenized strain. Gray dotted lines: Expression change and fitness of the un-mutagenized strain. Error bars: 95% confidence intervals of expression changes and fitness measures obtained from four replicate populations of each mutant. The black dotted line represents a LOESS regression of fitness on median expression with a smoothing parameter of 1% and 95% confidence intervals of the estimates shown as a gray shaded area.

allelic replacement. Sequencing the mutated regions of *RAP1* and *GCR1* in a random subset of transformants showed that each strain harbored an average of 1.8 mutations in the *RAP1* gene (*Figure 5C*) or 2.4 mutations in the *GCR1* gene (*Figure 5D*). As expected for PCR-based mutagenesis, the number of mutations per strain appeared to follow a Poisson distribution both for *RAP1* mutants (*Figure 5C*, Chi-square goodness of fit, p = 0.14) and *GCR1* mutants (*Figure 5D*, Chi-square goodness of fit, p = 0.79).

Among the *RAP1* mutant strains, only 9.1% (43 of 470 strains) showed a significant change in $P_{TDH3}$-*YFP* expression greater than 3% (corresponding to a ~1% change in fluorescence) relative to the un-mutagenized progenitor strain (*Figure 5E*), suggesting that most EMS mutants harboring coding mutations in *RAP1* would have been excluded from our mapping study. In addition, the strongest decrease in $P_{TDH3}$-*YFP* expression observed among *RAP1* mutants (17%) was substantially

smaller than the strongest decrease in expression caused by mutating the RAP1-binding site in the *TDH3* promoter (57.5% reported in *Duveau et al., 2018*), suggesting that even this most severe phenotype was not caused by a null allele of *RAP1*. To test this hypothesis, we used site-directed mutagenesis to alter five amino acids (one at a time) in Rap1p expected to disrupt DNA binding based on the crystal structure of Rap1p complexed with DNA (*Konig et al., 1996*). In each case, we obtained by PCR a DNA fragment containing either a synonymous mutation in the codon corresponding to the amino acid (which should not affect the DNA binding of Rap1p) or one of two nonsynonymous mutations, with one nonsynonymous mutation more likely to alter protein function than the other (*Yampolsky and Stoltzfus, 2005*). We then used CRISPR/Cas9 allele replacement to introduce each mutation into the yeast genome and sequenced 10 independent clones from each transformation to determine if the mutation was introduced in the *RAP1* coding sequence as intended. All five synonymous mutations were observed in several of the clones sequenced, but 7 of the 10 nonsynonymous mutations were never recovered (*Supplementary file 8*). This outcome suggests that nonsynonymous mutations altering the DNA binding of Rap1p are lethal or nearly lethal, making them unlikely to have been recovered in a mutagenesis screen. Indeed, Rap1p is known to be an essential, pleiotropic transcription factor playing critical roles in regulating expression of glycolytic genes like *TDH3* as well as ribosomal proteins and genes required for mating (reviewed in *Piña et al., 2003*). Taken together, these data indicate that *RAP1* mutations are unlikely to be common sources of variation in expression driven by the *TDH3* promoter.

For the *GCR1* mutant strains, 37.7% showed a significant change in $P_{TDH3}$-*YFP* expression greater than 3% relative to the un-mutagenized progenitor strain (*Figure 5F*). Several of these mutant alleles decreased the expression driven by the *TDH3* promoter by ~80%, which is similar to the previously reported effects of mutations in the Gcr1p binding sites of the *TDH3* promoter (*Metzger et al., 2015*), suggesting that they were null alleles. Indeed, resequencing these large effect alleles revealed that one of them had a single nucleotide insertion in the 28th codon of the *GCR1* ORF, which led to a frame shift eliminating 96% of amino acids (757 of 785) from Gcr1p. Because Gcr1p regulates expression of many glycolytic genes (*Uemura et al., 1997*) and *GCR1* deletion has been reported to cause severe growth defects in fermentable carbon source environments (*Clifton et al., 1978*; *Hossain et al., 2016*; *López and Baker, 2000*), we hypothesized that the fitness effects of mutations in *GCR1* might also have caused them to be underrepresented in the population from which the EMS mutants analyzed were derived. To test this hypothesis, we measured the relative fitness of 62 of the 220 *GCR1* mutants, including all mutants with decreased $P_{TDH3}$-*YFP* expression. *GCR1* mutants causing the largest changes in $P_{TDH3}$-*YFP* expression showed strong defects in growth rate; however, several *GCR1* mutants with changes in $P_{TDH3}$-*YFP* expression greater than 3% did not strongly affect fitness (*Figure 5G*). This observation suggests that some of the coding mutations in *GCR1* decreasing $P_{TDH3}$-*YFP* expression could have been sampled among the EMS mutants used for mapping. We therefore conclude that mutations in *GCR1* were most likely not recovered in our set of regulatory mutations because of the wide diversity of mutations that can affect *TDH3* expression and the limited number of EMS mutants included in the mapping experiment.

## Properties of genes harboring regulatory mutations

With only 5 of the 65 *trans*-regulatory point mutations in coding sequences mapping to transcription factors, we used gene ontology (GO) analysis to examine the types of genes harboring *trans*-regulatory mutations affecting $P_{TDH3}$-*YFP* expression more systematically. In all, these 65 mutations mapped to 42 different genes, with nine genes affected by more than one mutation, 4 of which were genes involved in the de novo purine biosynthesis pathway (*Figure 6A*). Several gene ontology terms were significantly enriched among genes affected by *trans*-regulatory mutations relative to genes affected by non-regulatory mutations. *Supplementary file 9* includes all enriched GO terms, whereas *Figure 6B* only includes enriched GO terms that are not parent to other GO terms in the GO hierarchy. Excluding mutations identified by sequencing candidate genes had a negligible impact on the outcome of the GO term analysis, with more than 96% of overlap between the GO terms found to be enriched before and after excluding mutations identified by Sanger sequencing (*Supplementary file 8*). Of the 33 GO terms enriched for *trans*-regulatory mutations shown in *Figures 6B,* 11 terms (including 13 of the 42 genes with *trans*-regulatory mutations) were related to chromatin structure (*Figure 6B*), which is known to play an important role in the regulation of gene expression (*Li et al., 2007*). An additional five GO terms (including six genes with *trans*-regulatory

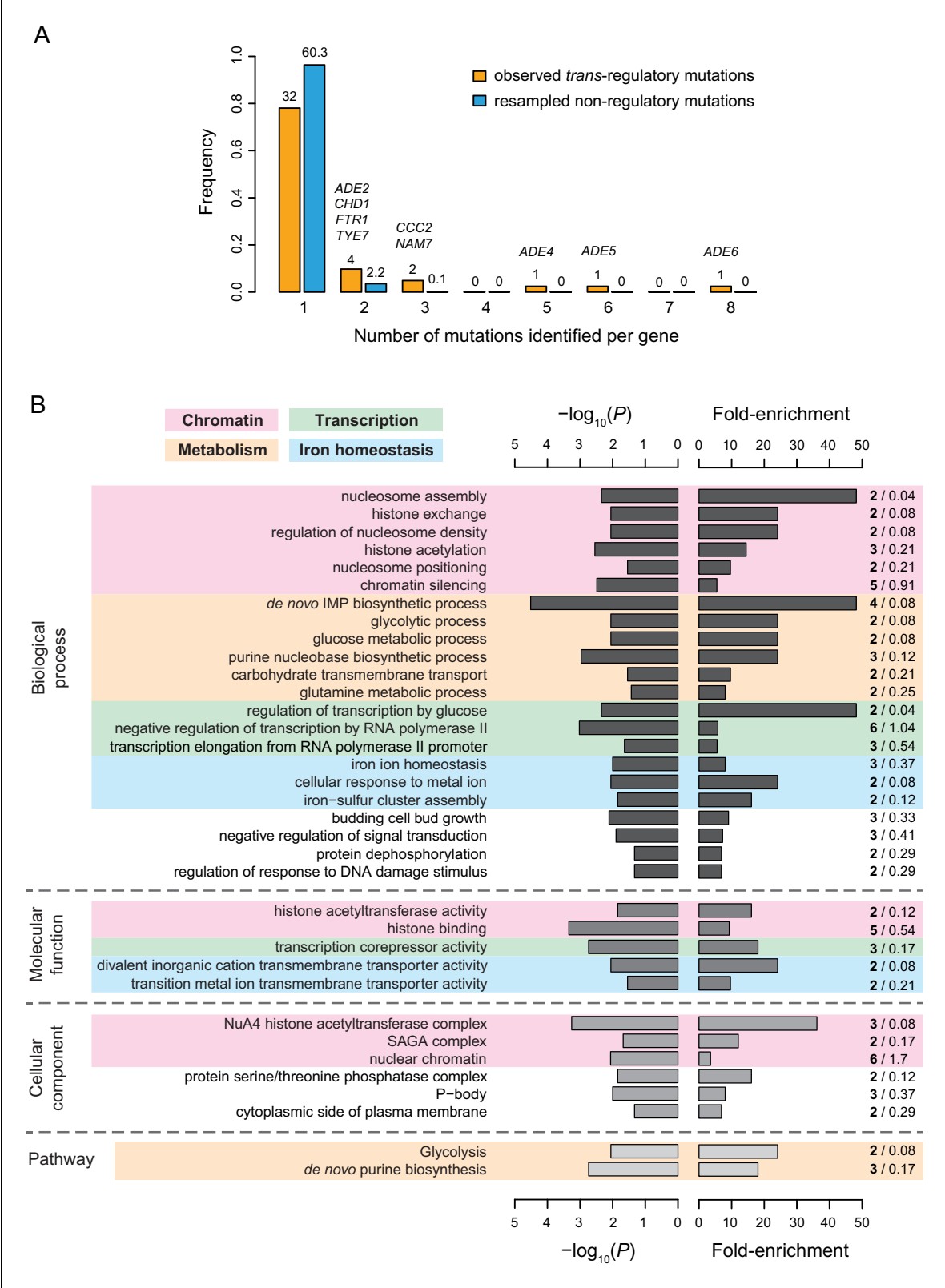

**Figure 6.** Properties of genes with coding mutations altering $P_{TDH3}$-*YFP* expression level. (**A**) Proportion of genes with one or more mutations identified among EMS mutants. Mutations in intergenic regions were excluded from this analysis. Orange bars include genes harboring one or more of the 65 *trans*-regulatory mutations identified in coding sequences. Blue bars include genes harboring one or more of 65 non-regulatory mutations randomly chosen among the set of 1095 non-regulatory mutations observed in coding sequences. The number of genes hit by 1–8 mutations is indicated above

*Figure 6 continued on next page*

*Figure 6 continued*

the corresponding bar. For blue bars, this number represents the mean number of genes obtained from 1000 random sets of 65 non-regulatory mutations. The names of genes with at least two *trans*-regulatory mutations identified among mutants are indicated above the bars. *FTR1* and *CCC2* are involved in iron homeostasis, *ADE2,4,5,6* are involved in de novo purine biosynthesis, *NAM7* is involved in nonsense-mediated mRNA decay, *CHD1* is involved in chromatin regulation and *TYE7* encodes a transcription factor regulating *TDH3* expression. (B) Summary of gene ontology (GO) enrichment analysis performed with PANTHER tool (http://www.pantherdb.org/). Fisher's exact tests were used to evaluate the overrepresentation of GO terms among the 42 genes affected by one or more of the 66 *trans*-regulatory mutations in coding sequences relative to the 1043 genes affected by one or more of the 1251 non-regulatory mutations in coding sequences. The descriptions shown on the left correspond to GO terms with a p-value < 0.05 (left bars), a fold-enrichment > 3 (right bars) and that are not parents to other GO terms in the ontology hierarchy (i.e. GO terms that are the most specific). A more complete list of enriched GO terms can be found in *Supplementary file 8*. Shades of gray represent different categories of GO terms (from darkest to lightest: biological processes, molecular functions and cellular components) or PANTHER pathways (lightest gray). Fold-enrichment was calculated as the observed number of genes with a particular GO term in the set of genes affected by *trans*-regulatory mutations (bold numbers on the right) divided by an expected number of genes obtained from the number of genes with the same GO term in the set of genes affected by non-regulatory mutations (regular numbers on the right). Four groups of GO terms and pathways involved in similar processes are represented by colored areas: chromatin (pink), metabolism (orange), transcription (green), and iron homeostasis (blue).

mutations) were related to metabolism, and four terms (including nine genes with *trans*-regulatory mutations) were related to transcriptional regulation (*Figure 6B*). Three GO terms related to glucose signaling, including regulation of transcription by glucose, carbohydrate transmembrane transport and glucose metabolic process, were also significantly enriched for genes affected by *trans*-regulatory mutations (*Figure 6B*). When we broadened this category of genes based on a review of glucose signaling (*Santangelo, 2006*), the enrichment included five genes implicated in glucose signaling (*Supplementary file 10*; 12.2% of genes affected by *trans*-regulatory mutations were involved in glucose signaling *vs* 2.7% of genes affected by non-regulatory mutations; Fisher's exact test: $p = 6.2 \times 10^{-3}$).

At the pathway level, we found that genes involved in glycolysis and de novo purine biosynthesis were also significantly enriched for *trans*-regulatory mutations (*Figure 6B*), with the latter driven by the mutations in *ADE2*, *ADE4*, *ADE5,* and *ADE6* genes described above (*Supplementary file 11*). Genes involved in iron homeostasis also emerged as an over-represented group, with five GO terms (including seven genes) being related to the regulation of intracellular iron concentration (*Figure 6B*). Diverse cellular processes implicated in iron homeostasis were represented among genes harboring *trans*-regulatory mutations, such as iron transport (*FTR1*, *CCC2*), iron trafficking and maturation of iron-sulfur proteins (*CIA2*, *NAR1*), transcriptional regulation of the iron regulon (*FRA1*), and post-transcriptional regulation of iron homeostasis (*TIS11*). Remarkably, nearly half of all *trans*-regulatory point mutations in coding sequences (31 of 65) were located in genes involved either in purine biosynthesis or iron homeostasis. Moreover, six of the eight genes harboring more than one *trans*-regulatory mutation (*Figure 6A*) were involved in one of these two processes. Mutations in purine biosynthesis genes tended to cause large increases in expression, whereas mutations in iron homeostasis genes tended to cause large decreases in expression (*Supplementary file 11*). Although the mechanistic relationship between these pathways and *TDH3* expression is not known, changing cellular conditions, including concentrations of metabolites (*Pinson et al., 2009*) or iron within the cell (reviewed in *Outten and Albetel, 2013*), can affect the regulation of gene expression. Ultimately, our data suggest that although mutations affecting $P_{TDH3}$-*YFP* expression map to genes with diverse functions, genes involved in a small number of well-defined biological processes are particularly likely to harbor such *trans*-regulatory mutations.

### *Trans*-regulatory mutations are enriched in genomic regions harboring natural variation affecting *TDH3* expression

Because new mutations affecting gene expression provide the raw material for regulatory variation segregating within a species, we asked whether the *trans*-regulatory mutations we observed were enriched in genomic regions associated with naturally occurring *trans*-regulatory variation affecting expression driven by the *TDH3* promoter. Specifically, we compared the genomic locations of *trans*-regulatory mutations identified in the current study to the locations of *trans*-acting quantitative trait loci (QTL) affecting expression of $P_{TDH3}$-*YFP* identified from crosses between the progenitor strain of

the EMS mutants (BY) and three other *S. cerevisiae* strains (SK1, YPS1000, M22) (*Metzger and Wittkopp, 2019*; *Figure 7A*).

Non-regulatory mutations were observed in eQTL regions as often as expected by chance (66.7% of non-regulatory mutations *vs* 65.1% of the whole genome in eQTL regions; *G*-test: p = 0.15), but the 66 *trans*-regulatory mutations were significantly enriched in eQTL regions (*Figure 7B*; 88% of *trans*-regulatory mutations *vs* 66.7% of non-regulatory mutations in eQTL regions; *G*-test: p = 9.6 x $10^{-5}$). The overrepresentation of *trans*-regulatory mutations in eQTL regions remained statistically

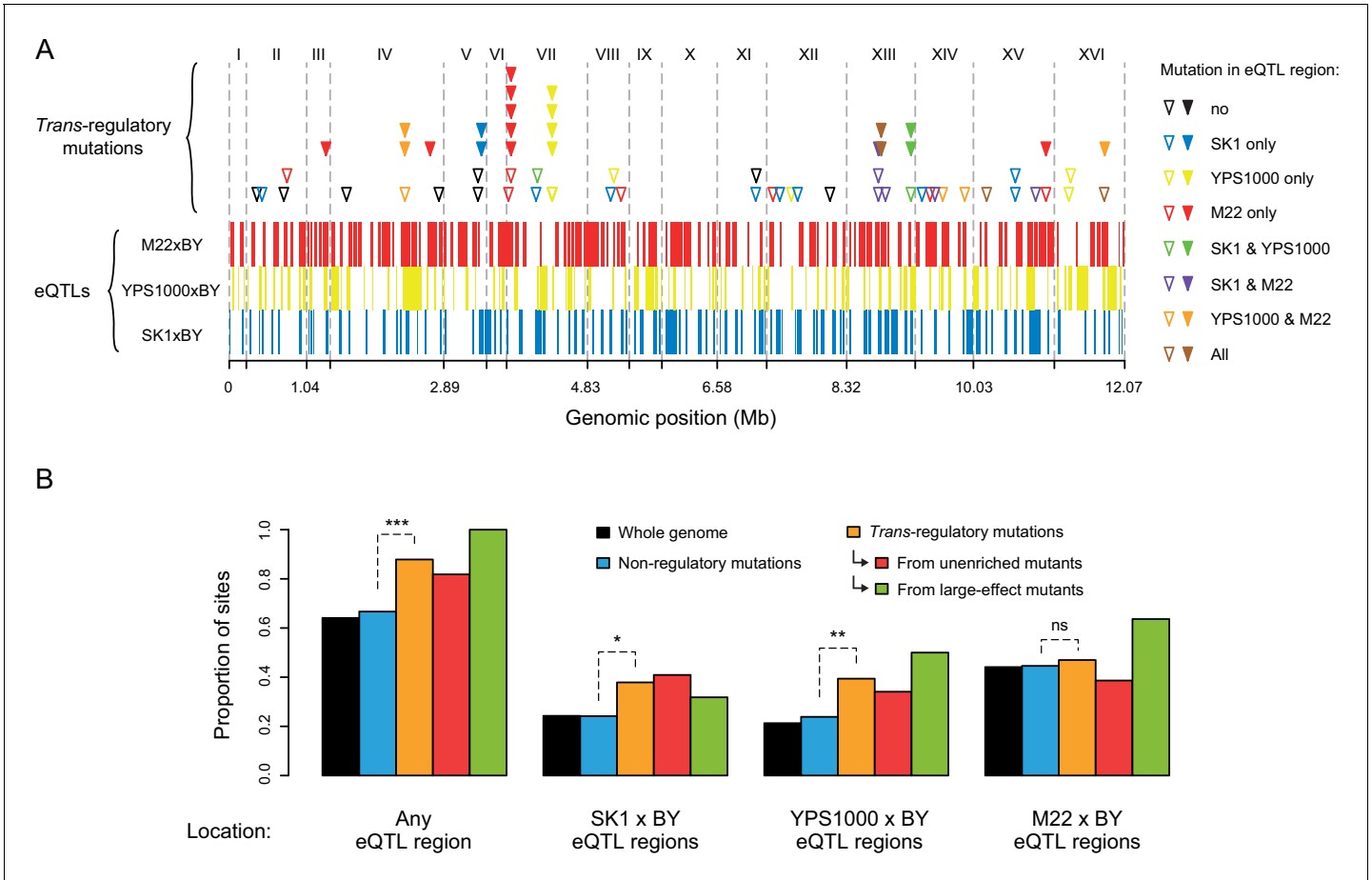

**Figure 7.** Overrepresentation of *trans*-regulatory mutations in eQTLs regions. (A) Overlap of 66 *trans*-regulatory point mutations and 317 eQTL regions along the yeast genome. eQTL regions were identified by BSA-Seq in *Metzger and Wittkopp, 2019* from three crosses of a laboratory strain (BY) to each of three strains expressing $P_{TDH3}$-YFP in the genetic background of different *S. cerevisiae* isolates: SK1 (eQTL regions represented by blue bars), YPS1000 (eQTL regions represented by yellow bars) and M22 (eQTL regions represented by red bars). Triangles indicate the genomic locations of *trans*-regulatory mutations, with open triangles representing mutations identified in mutants from the unenriched collection and filled triangles representing mutations identified in mutants enriched for large effects. Triangles are colored depending on the overlap between mutations and eQTL regions: black if the mutation is outside of any eQTL region, blue if the mutation lies in an eQTL region only identified from SK1xBY, yellow if the mutation lies in an eQTL region only identified from YPS1000xBY, red if the mutation lies in an eQTL region only identified from M22xBY, green if the mutation lies in two overlapping eQTL regions identified from SK1xBY and YPS1000xBY, purple if the mutation lies in two overlapping eQTL regions identified from SK1xBY and M22xBY, orange if the mutation lies in two overlapping eQTL regions identified from M22xBY and YPS1000xBY and brown if the mutation lies in three overlapping eQTL regions identified from the three crosses. (B) Proportions of non-regulatory and *trans*-regulatory mutations located in eQTL regions. Black bars: proportions of sites among the 12.07 Mb yeast genome. Blue bars: proportions of the 1759 non-regulatory point mutations. Orange bars: proportions of the 66 *trans*-regulatory mutations (excluding aneuploidies). Red bars: proportions of the 44 *trans*-regulatory mutations identified in mutants from the unenriched collection. Green bars: proportions of the 22 *trans*-regulatory mutations identified in mutants enriched for large effects. The proportions of non-regulatory and *trans*-regulatory mutations in eQTL regions were compared using *G*-tests (***: p < 0.001, **: 0.001 < p < 0.01, *: 0.01 < p < 0.05, ns: p > 0.05).

The online version of this article includes the following figure supplement(s) for figure 7:

**Figure supplement 1.** Proportions of different categories of non-regulatory mutations and *trans*-regulatory mutations located in eQTLs regions.

significant when we considered only the 44 *trans*-regulatory mutations identified from the collection of EMS mutants not enriched for large effects (*Figure 7B*; *G*-test: p = 0.027) or when we excluded the 17 *trans*-regulatory mutations identified by sequencing candidate genes (*Figure 7—figure supplement 1*; *G*-test: p = 8.4 x 10$^{-3}$). The enrichment of *trans*-regulatory mutations in eQTL regions was thus not driven solely by the effect size of these mutations or by the fact that several of the *trans*-regulatory mutations with large effects were located in the same genes. We also found that differences in sequencing coverage across the genome were unlikely to account for this enrichment (*Figure 7—figure supplement 1*). When we considered eQTL regions identified from each cross separately, we observed a significant enrichment of *trans*-regulatory mutations in eQTL regions identified in SK1 x BY and YPS1000 x BY crosses, but not in eQTL regions identified in the M22 x BY cross (*Figure 7B*; *G*-tests: p = 0.016 for SK1 x BY, p = 6.5 x 10$^{-3}$ for YPS1000 x BY, p = 0.70 for M22 x BY). Overall, the enrichment of *trans*-regulatory mutations in eQTL regions suggests that biases in the mutational sources of regulatory variation have shaped genetic sources of expression variation segregating in wild populations.

## Discussion

By systematically isolating and characterizing 69 *trans*-regulatory mutations that all affect expression of the same focal gene, this study reveals how *trans*-regulatory mutations are distributed within a genome and within a regulatory network. For example, we found that these *trans*-regulatory mutations were widely spread throughout the genome, with all except one located in coding sequences. These data also allowed us to determine how well a regulatory network inferred from integrating functional genomic and genetic data can predict sources of *trans*-regulatory variation. Like many biological networks, transcriptional regulatory networks have been inferred with the promise of explaining relationships between genetic variants and the higher order trait of gene expression, but the predictive power of such networks remains sparsely tested (*Flint and Ideker, 2019*).

We found that although the *trans*-regulatory mutations in coding regions were not enriched in transcription factors generally, they were overrepresented among transcription factors inferred to be regulators of *TDH3*. None of these transcription factors are known to directly bind to the *TDH3* promoter, however, and mutations in *RAP1* and *GCR1*, which have well characterized binding sites in the *TDH3* promoter, were notably missing from our set of *trans*-regulatory mutations affecting P$_{TDH3}$-*YFP* expression. Targeted mutagenesis of *RAP1* and *GCR1* suggested that most mutations in these genes (particularly *RAP1*) cause severe growth defects that might have prevented their recovery in mutagenesis screens. Over 90% of the *trans*-regulatory mutations examined were located in genes outside of this transcription factor network encoding proteins with diverse molecular functions involved in chromatin remodeling, nonsense-mediated mRNA decay, translation regulation, purine biosynthesis, iron homeostasis, and glucose sensing. Surprisingly, nearly half of the *trans*-regulatory mutations mapped to genes involved in either the purine biosynthesis or iron homeostasis pathways. Although not anticipated, finding so many *trans*-regulatory mutations in genes that are not transcription factors is consistent with the transcriptomic effects of gene deletions showing that transcription factors tend not to affect expression of more genes than other types of proteins (*Featherstone and Broadie, 2002*). Consequently, it seems that regulatory networks describing the relationships between transcription factors and target genes might capture only a small fraction of the potential sources of *trans*-regulatory variation.

Understanding the properties of *trans*-regulatory mutations is important because these mutations provide the raw material for natural *trans*-regulatory variation. We found that mutations affecting P$_{TDH3}$-*YFP* expression were enriched in genomic regions associated with expression variation among wild isolates of *S. cerevisiae*, suggesting that mutational sources of regulatory variation have had a lasting effect on sources of genetic variation affecting gene expression segregating in natural populations. This pattern is not necessarily expected if the *trans*-regulatory mutations we characterized captured only a small subset of the loci that can contribute to segregating *trans*-regulatory variation for this gene. Differences between the distribution of new *trans*-regulatory mutations and segregating *trans*-regulatory variants are also expected to arise when natural selection favors the maintenance of mutations at some loci more than others. Such differences in fitness can arise independently of a mutation's impact on *TDH3* expression because *trans*-acting mutations can also have pleiotropic effects on expression of other genes. A third reason why differences between the

mutational sources of *trans*-regulatory variation characterized here and *trans*-regulatory variation segregating in the wild can occur would be because of epistatic interactions among variants that are not captured by studying the effects of mutations individually. Ultimately, explaining the variation in gene expression we see in natural populations will require studies like this elucidating the mutational input as well as studies describing the fitness, pleiotropic, and epistatic effects of these mutations in native environments.

To the best of our knowledge this work provides the largest collection of individual mutations with *trans*-regulatory effects on expression of a single gene available to date, but it still only interrogates a single gene in a single species. Moreover, although the methods used were sensitive enough to identify genetic changes impacting expression of the focal gene as little as 1.6%, many mutations important for natural variation might have even smaller individual effects on a focal gene's expression and are thus missing from this study (*Rockman, 2012*). The chemical mutagen (EMS) used to generate the mutants analyzed in this work also captures only a subset of the type of mutations that arise naturally, and the use of a YFP reporter gene to measure activity of the *TDH3* promoter precluded recovery of *trans*-regulatory mutations that can impact native *TDH3* expression post-transcriptionally. The focal gene chosen for this work, *TDH3*, might also have properties that cause its spectrum of *trans*-regulatory mutations to differ from other genes in *S. cerevisiae*. For example, *TDH3* is one of the most highly expressed genes in *S. cerevisiae* (*Ghaemmaghami et al., 2003*), and it is one of the ~8% of genes in the *S. cerevisiae* genome that contains both a TATA box and a large nucleosome-free region in its promoter (*Tirosh and Barkai, 2008*). The metabolic functions of the TDH3p protein encoded by the *TDH3* gene might also cause its regulatory network to have properties that differ from genes encoding proteins with other types of functions (*Luscombe et al., 2004*).

It is tempting to extend these results from *S. cerevisiae* to other eukaryotes, but such extrapolation must take into account differences in genomes and gene regulatory mechanisms among species. For example, compared to species like fruit flies, mice, and humans, the baker's yeast *S. cerevisiae* has a much higher proportion of its genome (69.4%, https://www.yeastgenome.org/) that codes for proteins and much more compact *cis*-regulatory sequences (the median promoter length is 455 bp *Kristiansson et al., 2009*). Consequently, new *trans*-regulatory mutations in coding sequences might be more likely to arise in *S. cerevisiae* than in these other species. Most *S. cerevisiae* genes also lack introns (*Parenteau et al., 2019*) and DNA methylation is less prevalent than in many other eukaryotic species (*Tang et al., 2012*), so these potential sources of *trans*-regulatory variation in other species are unlikely to be captured when studying regulatory mutations in *S. cerevisiae*. Nonetheless, we think some observations, such as that genes with diverse functions can harbor *trans*-regulatory mutations, are likely to also apply to other eukaryotic species. Ultimately, we believe that this work provides an important foundation for understanding how the *trans*-regulatory mutations that give rise to *trans*-regulatory variation segregating in natural populations are structured within a genome and a regulatory network.

## Materials and methods

### Mutant strains selected for mapping

To identify mutations associated with expression changes, we selected 82 haploid mutant strains for bulk segregant analysis (*Figure 1A*) from three collections of mutants obtained in *Gruber et al., 2012* and *Metzger et al., 2016* via ethyl methanesulfonate (EMS) mutagenesis of two progenitor strains expressing a *YFP* reporter gene (Yellow Fluorescent Protein) under control of the *TDH3* promoter ($P_{TDH3}$-*YFP*). 71 mutants were selected from a collection of 1498 lines founded from cells isolated randomly (unenriched) after mutagenesis in *Metzger et al., 2016*, five mutants were selected from 211 lines founded from cells enriched for fluorescence changes after mutagenesis in *Metzger et al., 2016* and the last six mutants were selected from 1064 lines founded from cells enriched for fluorescence changes in *Gruber et al., 2012*. Mutants from *Metzger et al., 2016* were obtained by mutagenesis of the progenitor strain YPW1139 (*MATα ura3d0*), while mutants from *Gruber et al., 2012* were obtained by mutagenesis of the progenitor strain YPW1 (*MAT**a** ura3d0 lys2d0*). Both progenitors were derived from S288c genetic background (see *Metzger et al., 2016* and *Gruber et al., 2012* for details on construction of YPW1139 and YPW1 strains). In YPW1139,

$P_{TDH3}$-YFP is inserted at the *ho* locus with a *KanMX* drug resistance marker. In YPW1, $P_{TDH3}$-YFP is inserted at position 199270 on chromosome I near a pseudogene. YPW1139 harbors *RME1(ins-308A)* and *TAO3(1493Q)* alleles (*Deutschbauer and Davis, 2005*) that increase sporulation frequency relative to YPW1 alleles, as well as *SAL1*, *CAT5* and *MIP1* alleles that decrease the frequency of the petite phenotype (*Dimitrov et al., 2009*). We previously showed that the few genetic differences between YPW1 and YPW1139 did not affect the magnitude of effects of *TDH3* promoter mutations on fluorescence (*Metzger et al., 2016*). Fluorescence levels of the three collections were measured in *Gruber et al., 2012* and in *Metzger et al., 2016*. From these data, we selected 39 mutants for BSA-Seq that showed statistically significant fluorescence changes greater than 1% relative to the progenitor strain. Among these mutants, six were selected from the *Gruber et al., 2012* collection (Z-score > 2.58, p < 0.01), five were selected from mutants enriched for large effects in *Metzger et al., 2016* (permutation test, p < 0.05) and 28 were selected from unenriched mutants in *Metzger et al., 2016* (permutation test, p < 0.05). The remaining 43 mutants included in BSA-Seq experiments were selected from mutants in *Metzger et al., 2016* for which we collected new fluorescence measures using flow cytometry. This second fluorescence screen included 197 lines from the unenriched collection that were chosen because they showed statistically significant fluorescence changes (permutation test, p < 0.05) greater than 1% relative to the progenitor strain in the initial screen published in *Metzger et al., 2016*. The 43 mutants selected from this 2nd screen showed statistically significant fluorescence changes (permutation test, p < 0.05) greater than 1% relative to the progenitor strain.

## Measuring YFP expression by flow cytometry

Fluorescence levels of mutant strains were quantified by flow cytometry using the same approach as described in *Metzger et al., 2016* and *Duveau et al., 2018*. For assays involving strains stored in individual tubes at −80°C, all strains were thawed in parallel on YPG plates (10 g yeast extract, 20 g peptone, 50 ml glycerol, 20 g agar per liter) and grown for 2 days at 30°C. Strains were then arrayed using pipette tips in 96 deep well plates containing 0.5 ml of YPD medium (10 g yeast extract, 20 g peptone, 20 g D-glucose per liter) per well at positions defined in *Supplementary file 12*. The reference strain YPW1139 was inoculated at 20 fixed positions on each plate to correct for plate and position effects on fluorescence. The non-fluorescent strain YPW978 was inoculated in one well per plate to quantify the autofluorescence of yeast cells. Plates were incubated at 30°C for 20 hr with 250 rpm orbital shaking (each well contained a sterile 3 mm glass bead to maintain cells in suspension). Samples from each plate were then transferred to omnitrays containing YPG-agar using a V&P Scientific pin tool. For assays involving strains already arrayed in 96-well plates at −80°C (i.e. *RAP1* and *GCR1* mutants), strains were directly transferred on YPG omnitrays after thawing. After 48 hr of incubation at 30°C, samples from each omnitray were inoculated using the pin tool in four replicate 96-well plates containing 0.5 ml of YPD per well and cultivated at 30°C with 250 rpm shaking for 22 hr. Then, 15 µl of cell cultures were transferred to a 96-well plate with 0.5 ml of PBS per well (phosphate-buffered saline) and samples were immediately analyzed on a BD Accuri C6 flow cytometer connected to a HyperCyt autosampler (IntelliCyt Corp). A 488 nm laser was used for excitation and the YFP signal was acquired with a 530/30 optical filter. Each well was sampled for 2 s, yielding fluorescence and cell size measurements for at least 5000 events per well. Flow cytometry data were analyzed using custom R scripts (*Source code 1*) as described in *Duveau et al., 2018*. First, events that did not correspond to single cells were filtered out using *flowClust* clustering functions. Second, fluorescence intensity was scaled by cell size in several steps. For *Figure 1B–D*, these values of fluorescence relative to cell size were directly used for subsequent steps of the analysis. For other figures, these values were transformed using a log-linear function to be linearly related with YFP abundance. Transformations of fluorescence values were performed using the relationship between fluorescence levels and *YFP* mRNA levels established in *Duveau et al., 2018* from five strains carrying mutations in the promoter of the $P_{TDH3}$-YFP reporter gene. The YFP mRNA levels quantified in these five strains are expected to be linearly related with YFP protein abundance based on a previous study that compared mRNA and protein levels for a similar fluorescent protein (GFP) across a broad range of expression levels (*Kafri et al., 2016*). For this reason and because mutations recovered in this study may alter YFP expression at the post-transcriptional level, the transformed values of fluorescence were considered to provide estimates of YFP abundance instead of mRNA levels. The median expression among all cells of each sample was then corrected to account for positional

effects estimated from a linear model applied to the median expression of the 20 control samples on each plate. To correct for autofluorescence, the mean of median expression measured among all replicate populations of the non-fluorescent strain was then subtracted from the median expression of each sample. Finally, a relative measure of expression was calculated by dividing the median expression of each sample by the mean of the median expression among replicates of the reference strain. Samples for which the relative expression differed from the median expression among replicate populations by more than five times the median absolute deviation measured among replicate populations were considered as outliers and ignored. Figures show the mean relative expression among replicate populations of each genotype. Permutation tests used to compare the expression level of each single site mutant to the expression level of the EMS mutant carrying the same mutation are described in the legend of *Figure 2—figure supplement 5A*.

## Two-level permutation tests

We developed a permutation-based approach to determine which EMS mutant strains from *Metzger et al., 2016* showed a significant change in YFP expression relative to their progenitor strain. This permutation approach was motivated by the fact that Student tests and Mann-Whitney-Wilcoxon tests applied to these data appeared to be overpowered. Indeed, the flow cytometry assay from *Metzger et al., 2016* included 146 instances of the progenitor strain YPW1139 that were placed at random plate positions and with fluorescence measured in four replicate populations for each position. When comparing the mean expression of the four replicate populations of YPW1139 grown at a given plate position to the mean expression of all other replicate populations of YPW1139, the p-value was below 0.05 in 25.3% of cases when using Student tests and in 13.7% of cases when using Mann-Whitney-Wilcoxon tests. The fact that more than 5% of p-values were below 0.05 indicated that the tests were overpowered, which was because expression differences between YPW1139 populations grown at different plate positions were in average larger than expression differences between replicate populations grown at the same position. For this reason, we compared the expression of each mutant strain to the expression of the 146 x 4 populations of the YPW1139 progenitor strain using permutation tests with two levels of resampling as described below. In these tests, we compared 10,000 times the expression levels of each tested strain measured in quadruplicates to the expression levels of YPW1139 measured in quadruplicates at a randomly selected plate position among the 146 available positions (a new position was picked at each iteration). For each iteration of the comparison, we calculated the difference $D$ between (1) the absolute difference observed between the mean expression of the tested strain and the mean expression of YPW1139 and (2) a randomized absolute difference of mean expression between two sets of four expression values obtained by random permutation of the four expression values measured for the tested strain and of the four expression values measured for YPW1139 at the selected plate position. Finally, for each tested strain the proportion of $D$ values that were negative (after excluding $D$ values equal to zero) corresponded to the p-value of the permutation test. When we applied this test to YPW1139 as a tested strain, we found that the p-value was below 0.05 for 6.1% of the 146 plate positions containing YPW1139, indicating that the permutation test was not overpowered.

## BSA-Seq procedure

To identify mutations associated with fluorescence levels in EMS-treated mutants, we used bulk-segregant analysis followed by Illumina sequencing (BSA-Seq). BSA-Seq data corresponding to the six mutants from *Gruber et al., 2012* were collected together with the BSA-Seq dataset published in *Duveau et al., 2014*. For the other 76 mutants (from *Metzger et al., 2016*), BSA-Seq data were collected in this study in several batches (see *Supplementary file 13*) using the experimental approach described in *Duveau et al., 2014* (with few modifications). First, each EMS-treated mutant (*MATα ura3d0 ho::P_TDH3_-YFP ho::KanMX*) was crossed to the mapping strain YPW1240 (*MATa ura3d0 ho::P_TDH3_-YFP ho::NatMX4 mata2::yEmRFP-HygMX*) that contained the *FASTER MT* system from *Chin et al., 2012* used to tag diploid and *MATa* cells with a fluorescent reporter. Crosses were performed on YPD agar plates and replica-plated on YPD + G418 + Nat medium (YPD agar with 350 mg/L geneticin (G418) and 100 mg/L Nourseothricin) to select diploid hybrids. After growth, cells were streaked on another YPD + G418 + Nat agar plate, one colony was patched on YPG agar for each mutant and the diploid strain was kept frozen at −80℃. Bulk segregant populations were then

collected for batches of eight mutants in parallel as follows. Diploid strains were thawed and revived on YPG plates, grown for 12 hr at 30°C on GNA plates (50 g D-glucose, 30 g Difco nutrient broth, 10 g yeast extract and 20 g agar per liter) and sporulation was induced for 4 days at room temperature on KAc plates (10 g potassium acetate and 20 g agar per liter). For each mutant, we then isolated a large population of random spores (> $10^8$ spores) by digesting tetrads with zymolyase, vortexing, and sonicating samples in 0.02% triton-X (exactly as described in *Duveau et al., 2014*). ~3 x $10^5$ *MATα* spores were sorted by FACS (BD FACSAria II) based on the absence of RFP fluorescence signal measured using a 561 nm laser and 582/15 optical filter. Spores were then resuspended in 2 ml of YPD medium. After 24 hr of growth at 30°C, 0.4 ml of cell culture was transferred to a 5 ml tube containing 2 ml of PBS. Three populations of 1.5 x $10^5$ segregant cells were then collected by FACS: (1) a low fluorescence population of cells sorted among the 2.5% of cells with lowest fluorescence levels ('low bulk'), (2) a high fluorescence population of cells sorted among the 2.5% of cells with highest fluorescence levels ('high bulk'), and (3) a control population of cells sorted regardless of their fluorescence levels. YFP signal was measured using a 488 nm laser and a 530/30 optical filter. To exclude budding cells and enrich for single cells, ~70% of all events were filtered out based on the area and width of the forward scatter signal prior to sorting. In addition, the median FSC.A (area of forward scatter, a proxy for cell size) was maintained to similar values in the low fluorescence bulk and in the high fluorescence bulk by drawing sorting gates that were parallel to the linear relationship between FSC.A and fluorescence intensity in the FACSDiva software. After sorting, cells were resuspended in 1.6 ml of YPD medium and grown for 30 hr at 30°C. Each sample was then stored at −80°C in 15% glycerol in two separate tubes: one tube containing 1 ml of culture (for DNA extraction) and one tube containing 0.5 ml of culture (for long-term storage). Extraction of genomic DNA was performed for 24 samples in parallel using a Gentra Puregene Yeast/Bact kit (Qiagen). Then, DNA libraries were prepared from 1 ng of genomic DNA using Nextera XT DNA Library Prep kits (Illumina) for low fluorescence bulks and for high fluorescence bulks (control populations were not sequenced). Tagmentation was carried out at 55°C for 5 min. Dual indexing of the libraries was achieved using index adapters provided in the Nextera XT Index kit (index sequences used for each library are indicated in *Supplementary file 14*). Final library purification and size selection was achieved using Agencourt AMPure XP beads (30 µl of beads added to 50 µl of PCR-amplified libraries followed by ethanol washes and resuspension in 50 µl of Tris-EDTA buffer). The average size of DNA fragments in the final libraries was 650 bp, as quantified from a subset of samples using high sensitivity assays on a 2100 Bioanalyzer (Agilent). The concentration of all libraries was quantified with a Qubit 2.0 Fluorometer (Thermo Fisher Scientific) using dsDNA high sensitivity assays. Libraries to be sequenced in the same flow lane were pooled to equal concentration in a single tube and sequenced on a HiSeq4000 instrument (Illumina) at the University of Michigan Sequencing Core Facility (150 bp paired-end sequencing). The 2 x 76 libraries were sequenced in four distinct sequencing runs (45300, 45301, 54374 and 54375) that included 36–54 samples (libraries sequenced in each run are indicated in *Supplementary file 14*). In addition, four control libraries were sequenced in run 45300, corresponding to genomic DNA from (1) YPW1139 progenitor strain, (2) YPW1240 mapping strain, (3) a bulk of low fluorescence segregants from YPW1139 x YPW1240 cross, and (4) a bulk of high fluorescence segregants from YPW1139 x YPW1240 cross. 18 libraries sequenced in run 54374 were not analyzed in this study.

## Analysis of BSA-Seq data

Demultiplexing of sequencing reads and generation of FASTQ files were performed using Illumina *bcl2fastq* v1.8.4 for sequencing runs 45300 and 45301 and *bcl2fastq2* v2.17 for runs 54374 and 54375. The next steps of the analysis were processed on the Flux cluster administered by the Advanced Research Computing Technology Services of the University of Michigan (script available in *Source code 4*). First, low quality ends of reads were trimmed with *sickle* (https://github.com/najoshi/sickle; *Joshi and Fass, 2011*) and adapter sequences were removed with *cutadapt* (*Martin, 2011*). Reads were then aligned to the S288c reference genome (https://www.yeastgenome.org/, R64-1-1 release to which we added the sequences corresponding to $P_{TDH3}$-*YFP*, *KanMX*, and *NatMX4* transgenes, available in *Supplementary file 12*) using *bowtie2* (*Langmead and Salzberg, 2012*) and overlaps between paired reads were clipped using *clipOverlap* in *bamUtil* (https://github.com/statgen/bamUtil). The sequencing depth at each position in the genome was determined using *bedtools genomecov* (https://github.com/arq5x/bedtools2). For variant calling, BAM files

corresponding to the low fluorescence bulk and to the high fluorescence bulk of each mutant were processed together using *freebayes* (https://github.com/ekg/freebayes; *Garrison and Marth, 2012*) with options `--pooled-discrete --pooled-continuous`. That way, sequencing data from both bulks were pooled to increase the sensitivity of variant calling and allele counts were reported separately for each bulk. To obtain a list of mutations present in each mutant strain, false positive calls in the VCF files generated by *freebayes* were then filtered out with the Bioconductor package *VariantAnnotation* in R (*Source code 2*). Filtering was based on the values of several parameters such as quality of genotype inference (QUAL > 200), mapping quality (MQM > 27), sequencing depth (DP > 20), counts of reference and alternate alleles (AO > three and RO > 3), frequency of the reference allele (FREQ.REF > 0.1), proportion of reference and alternate alleles supported by properly paired reads (PAIRED > 0.8 and PAIREDR > 0.8), probability to observe the alternate allele on both strands (SAP < 100) and at different positions of the reads (EPP < 50 and RPP < 50). The values of these parameters were chosen to filter out a maximum number of calls while retaining 28 variants previously confirmed by Sanger sequencing. We then used likelihood ratio tests (*G*-tests) in R to determine for each variant site whether the frequency of the alternate allele (i.e. the mutation) was statistically different between the low fluorescence bulk and the high fluorescence bulk (*Source code 2*). A point mutation was considered to be associated with fluorescence (directly or by linkage) if the p-value of the *G*-test was below 0.001, corresponding to a *G* value above 10.828. Since this *G*-test was performed for a total of 1819 mutations, we expected that 1.82 mutations would be associated with fluorescence due to type I error (false positives) at a p-value threshold of 0.001. This expected number of false positives was considered acceptable since it represented only 2.7% of all mutations that were associated with fluorescence. To determine if an aneuploidy was associated with fluorescence level, we compared the sequencing coverage of the aneuploid chromosome to genome-wide sequencing coverage in the low and high fluorescence bulks using *G*-tests. The *G* statistics was computed from the number of reads mapping to the aneuploid chromosome and the number of reads mapping to the rest of the genome in the low and high fluorescence bulks. Aneuploidies with *G* > 10.828, which corresponds to p-value < 0.001, were considered to be present at statistically different frequencies in both bulks. A custom R script was used to annotate all mutations identified in BSA-Seq data (*Source code 3*), retrieving information about the location of mutations in intergenic, intronic or exonic regions, the name of genes affected by coding mutations or the name of neighboring genes in case of intergenic mutations and the expected impact on amino acid sequences (synonymous, nonsynonymous, or nonsense mutation and identity of the new amino acid in case of a nonsynonymous mutation).

## Sanger sequencing of candidate genes

As an alternative approach to BSA-Seq, additional mutations were identified by directly sequencing candidate genes in a subset of EMS-treated mutants (*Supplementary file 4*). More specifically, we sequenced the $P_{TDH3}$-YFP transgene in 95 mutant strains from *Metzger et al., 2016* that showed decreased fluorescence by more than 10% relative to the progenitor strain. We sequenced the *ADE4* coding sequence in 14 mutants from *Metzger et al., 2016* that were not included in the BSA-Seq assays and that showed increased fluorescence by more than 5% relative to the progenitor strain. Two of the sequenced mutants had a mutation in the *ADE4* coding sequence. We then sequenced the *ADE5* coding sequence in the remaining 12 mutants and found a mutation in five of the sequenced mutants. We continued by sequencing the *ADE6* coding sequence in the remaining seven mutants. Five of the sequenced mutants had a single mutation and one mutant had two mutations in the *ADE6* coding sequence. We sequenced the *ADE8* coding sequence in the last mutant but we found no candidate mutation in this mutant. Finally, we sequenced the *ADE2* coding sequence in two mutants that showed a reddish color when growing on YPD plates. For all genes, the sequenced region was amplified by PCR from cell lysates, PCR products were cleaned up using Exo-AP treatment (7.5 µl PCR product mixed with 0.5 µl Exonuclease-I (NEB), 0.5 µl Antarctic Phosphatase (NEB), 1 µl Antarctic Phosphastase buffer and 0.5 µl $H_2O$ incubated at 37°C for 15 min followed by 80°C for 15 min) and Sanger sequencing was performed by the University of Michigan Sequencing Core Facility. Oligonucleotides used for PCR amplification and sequencing are indicated in *Supplementary file 14*.

## Site-directed mutagenesis

Thirty-four mutations identified by BSA-Seq and 11 mutations identified by sequencing candidate genes were introduced individually in the genome of the progenitor strain YPW1139 to quantify the effect of these mutations on fluorescence level. 'Scarless' genome editing (i.e. without insertion of a selection marker) was achieved using either the delitto perfetto approach from *Stuckey et al., 2011* (for 19 mutations) or CRISPR-Cas9 approaches derived from *Laughery et al., 2015* (for 26 mutations). Compared to delitto perfetto, CRISPR-Cas9 is more efficient and it can be used to introduce mutations in essential genes. However, it requires specific sequences in the vicinity of the target mutation (see below). The technique used for the insertion of each mutation is indicated in *Supplementary file 15*. The sequences of oligonucleotides used for the insertion and the validation of each mutation can be found in *Supplementary file 14*.

In the delitto perfetto approach, the target site was first replaced by a cassette containing the *Ura3* and *hphMX4* selection markers (pop-in) and then this cassette was swapped with the target mutation (pop-out). The *Ura3-hphMX4* cassette was amplified from pCORE-UH plasmid using two oligonucleotides that contained at their 5' end 20 nucleotides for PCR priming in pCORE-UH and at their 3' end 40 nucleotides corresponding to the sequences flanking the target site in the yeast genome (for homologous recombination). The amplicon was transformed into YPW1139 cells using a classic LiAc/polyethylene glycol heat shock protocol (*Gietz and Schiestl, 2007*). Cells were then plated on synthetic complete medium lacking uracil (SC-Ura) and incubated for two days at 30℃. Colonies were replica-plated on YPD + Hygromycin B (300 mg/l) plates. A dozen [Ura+ Hyg+] colonies were streaked on SC-Ura plates to remove residual parental cells and the resulting colonies were patched on YPG plates to counterselect petite cells. Cell patches were then screened by PCR to confirm the proper insertion of *Ura3-hphMX4* at the target site. One positive clone was grown in YPD and stored at −80℃ in 15% glycerol. For the pop-out step, a genomic region of ~240 bp centered on the mutation was amplified from the EMS-treated mutant containing the desired mutation. The amplicon was transformed into the strain with *Ura3-hphMX4* inserted at the target site. Cells were plated on a synthetic complete medium containing 0.9 g/l of 5-fluoroorotic acid (SC + 5-FOA) to counterselect cells expressing *Ura3*. After growth, a dozen [Ura-] colonies were streaked on SC + 5-FOA plates and one colony from each streak was patched on a YPG plate. Cell patches were screened by PCR using oligonucleotides that flanked the sequence of the transformed region and amplicons of expected size (~350 bp) were sequenced to confirm the insertion of the desired mutation and the absence of PCR-induced mutations. When possible two independent clones were stored at −80℃ in 15% glycerol, but in some cases only one positive clone could be retrieved and stored.

A 'one-step' CRISPR-Cas9 approach was used to insert mutations impairing a NGG or CCN motif in the genome (22 mutations), which corresponds to the protospacer adjacent motif (PAM) targeted by Cas9. First, a DNA fragment containing the 20 bp sequence upstream of the target PAM in the yeast genome was cloned between SwaI and BclI restriction sites in the pML104 plasmid. This DNA fragment was obtained by hybridizing two oligonucleotides designed as described in *Laughery et al., 2015*. The resulting plasmid contained cassettes for expression of Ura3, Cas9 and a guide RNA targeted to the mutation site in yeast cells. In parallel, a repair fragment containing the mutation was obtained either by PCR amplification of a ~240 bp genomic region centered on the mutation in the EMS-treated mutant or by hybridization of two complementary 70-mer oligonucleotides containing the mutation and its flanking genomic sequences. The Cas9/sgRNA plasmid and the repair fragments were transformed together (~150 nmol of plasmid + 20 µmol of repair fragment) into the progenitor strain YPW1139 using LiAc/polyethylene glycol heat shock protocol (*Gietz and Schiestl, 2007*). Cells were then plated on SC-Ura medium and incubated at 30℃ for 48 hr. This medium selected cells that both internalized the plasmid and integrated the desired mutation in their genome. Indeed, cells with the Cas9/sgRNA plasmid stop growing as long as their genomic DNA is cleaved by Cas9 but their growth can resume once the PAM sequence is impaired by the mutation, which is integrated into the genome via homologous recombination with the repair fragment (*Laughery et al., 2015*). A dozen [Ura+] colonies were then streaked on SC-Ura plates and one colony from each streak was patched on a YPG plate. Cell patches were screened by PCR using oligonucleotides that flanked the mutation site and amplicons of expected size (~350 bp) were sequenced to confirm the insertion of the desired mutation and the absence of secondary mutations.

Then, one or two positive clones were patched on SC + 5-FOA to counterselect the Cas9/sgRNA plasmid, grown in YPD and stored at −80℃ in 15% glycerol.

A 'two-steps' CRISPR-Cas9 approach was used to insert mutations located near but outside a PAM sequence (four mutations). Each step was performed as described above for the 'one-step' CRISPR-Cas9 approach. In the first step, Cas9 was targeted by the sgRNA to a PAM sequence (the initial PAM) located close to the mutation site (up to 20 bp). The repair fragment contained two synonymous mutations that were not the target mutation: one mutation that impaired the initial PAM and one mutation that introduced a new PAM as close as possible to the target site. This repair fragment was obtained by hybridization of two complementary 90-mer oligonucleotides and transformed into YPW1139. In the second step, Cas9 was targeted to the new PAM. The repair fragment contained three mutations: two mutations that reverted the mutations introduced in the first step and the target mutation. This repair fragment was obtained by hybridization of two complementary 90-mer oligonucleotides and transformed into the strain obtained in the first step. Positive clones were sequenced to confirm the insertion of the target mutation and the absence of other mutations.

We used CRISPR/Cas9-guided allele replacement to introduce individual mutations in five codons of the *RAP1* coding sequence that encode for amino acids predicted to make direct contact with DNA when RAP1 binds to DNA (*Konig et al., 1996*). For each codon, we tried to insert one synonymous mutation, one nonsynonymous mutation predicted to have a weak impact on RAP1 protein structure and one nonsynonymous mutation predicted to have a strong impact on RAP1 protein structure based on amino acid exchangeability scores from *Yampolsky and Stoltzfus, 2005* (see *Supplementary file 8* for the list of mutations). Each mutation was introduced in the genome of strain YPW2706. This strain is derived from YPW1139 and contains two identical sgRNA target sites upstream and downstream of the *RAP1* gene (see below for details on YPW2706 construction). Therefore, we could use a single Cas9/sgRNA plasmid to excise the entire *RAP1* gene in YPW2706 by targeting Cas9 to both ends of the gene. We used gene SOEing (Splicing by Overlap Extension) to generate repair fragments corresponding to the *RAP1* gene (promoter and coding sequence) with each target mutation. First, a left fragment of *RAP1* was amplified from YPW1139 genomic DNA using a forward 20-mer oligonucleotide priming upstream of the RAP1 promoter and a reverse 60-mer oligonucleotide containing the target mutation and the surrounding *RAP1* sequence. In parallel, a right fragment of RAP1 overlapping with the right fragment was amplified from YPW1139 genomic DNA using a forward 60-mer oligonucleotide complementary to the reverse oligonucleotide used to amplify the left fragment and a reverse 20-mer oligonucleotide priming in *RAP1* 5'UTR sequence. Then, equimolar amounts of the left and right fragments were mixed in a PCR reaction and 25 cycles of PCR were performed to fuse both fragments. Finally, the resulting product was further amplified using two 90-mer oligonucleotides with homology to the sequence upstream of *RAP1* promoter and to the *RAP1* 5'UTR but without the sgRNA target sequences. Consequently, transformation of the repair fragment together with the Cas9/sgRNA plasmid in YPW2706 cells was expected to replace the wild type allele of *RAP1* by an allele containing the target mutation in *RAP1* coding sequence and without the two flanking sgRNA target sites. For each of the 15 target mutations, we sequenced the *RAP1* promoter and coding sequence in 10 independent clones obtained after transformation. All synonymous mutations were retrieved in several clones, while several of the nonsynonymous mutations were not found in any clone, suggesting they were lethal (*Supplementary file 8*).

### *RAP1* and *GCR1* mutagenesis using error-prone PCR

We used a mutagenic PCR approach to efficiently generate hundreds of mutants with random mutations in the *RAP1* gene (promoter and coding sequence) or in the second exon of *GCR1* (representing 99.7% of *GCR1* coding sequence). DNA fragments obtained from the mutagenic PCR were introduced in the yeast genome using CRISPR/Cas9-guided allele replacement as described above. The sequences of all oligonucleotides used for *RAP1* and *GCR1* mutagenesis can be found in *Supplementary file 14*.

First, we constructed two yeast strains for which the *RAP1* gene (strain YPW2706) or the second exon of *GCR1* (strain YPW3082) were flanked by identical sgRNA target sites and PAM sequences. To generate strain YPW2706, we first identified a sgRNA target site located downstream of the *RAP1* coding sequence (41 bp after the stop codon in the 5'UTR) in the S288c genome. Then, we inserted the 23 bp sequence corresponding to this sgRNA target site and PAM upstream of the

*RAP1* promoter (immediately after *PPN2* stop codon) in strain YPW1139 using the delitto perfetto approach (as described above). To generate strain YPW3082, we first identified a sgRNA target site located at the end of the *GCR1* intron (22 bp upstream of exon 2) in the S288c genome. Then, we inserted the 23 bp sequence corresponding to this sgRNA target site and PAM immediately after the *GCR1* stop codon in strain YPW1139 using the delitto perfetto approach (as described above).

Second, we constructed plasmid pPW437 by cloning the 20mer guide sequence directed to *RAP1* in pML104 as described in *Laughery et al., 2015* and we constructed plasmid pPW438 by cloning the 20mer guide sequence directed to *GCR1* in pML104 as described in *Laughery et al., 2015*. These two sgRNA/Cas9 plasmids can be used, respectively, to excise the *RAP1* gene or *GCR1* exon two from the genomes of YPW2706 and YPW3082.

Third, we generated repair fragments with random mutations in *RAP1* or *GCR1* genes using error-prone PCR. We first amplified each gene from 2 ng of YPW1139 genomic DNA using a high-fidelity polymerase (KAPA HiFi DNA polymerase) and 30 cycles of PCR. PCR products were purified with the Wizard SV Gel and PCR Clean-Up System (Promega) and quantified with a Qubit 2.0 Fluorometer (Thermo Fisher Scientific) using dsDNA broad range assays. Two nanograms of purified PCR products were used as template for a first round of mutagenic PCR and mixed with 25 µl of Dream-Taq Master Mix 2x (ThermoFisher Scientific), 2.5 µl of forward and reverse primers at 10 µM, 5 µl of 1 mM dATP and 5 µl of 1 mM dTTP in a final volume of 50 µl. The imbalance of dNTP concentrations (0.3 µM dATP, 0.2 µM dCTP, 0.2 µM dGTP and 0.3 µM dTTP) was done to bias the mutagenesis toward misincorporation of dATP and dTTP. For *RAP1* mutagenesis, the forward oligonucleotide primed upstream of the *RAP1* promoter (in *PPN2* coding sequence) and the reverse oligonucleotide primed in the *RAP1* terminator and contained a mutation in the PAM adjacent to the sgRNA target site. For *GCR1* mutagenesis, the forward oligonucleotide primed at the end of the *GCR1* intron and contained a mutation in the PAM adjacent to the sgRNA target site and the reverse primer primed in the *GCR1* terminator. The PCR program was 95℃ for 3 min followed by 32 cycles with 95℃ for 30 s, 52℃ for 30 s, 72℃ for 2 min and a final extension at 72℃ for 5 min. For *RAP1* mutagenesis, the product of the first mutagenic PCR was diluted by a factor of 33 and used as template for a second round of mutagenic PCR (1.5 µl of product in a 50 µl reaction) similar to the first round but with only 10 cycles of amplification. For *GCR1* mutagenesis, the product of the first mutagenic PCR was diluted by a factor of 23 and used as template for a second round of mutagenic PCR (2.2 µl of product in a 50 µl reaction) with 35 cycles of amplification. Using this protocol, we expected to obtain on average 1.6 mutations per fragment for *RAP1* mutagenesis and 1.8 mutations per fragment for *GCR1* mutagenesis (see below for calculations of these estimates).

pPW437 was transformed with *RAP1* repair fragments into YPW2706 and pPW438 was transformed with *GCR1* repair fragments into YPW3082 as described above for CRISPR/Cas9 site directed mutagenesis. To select cells that replaced the wild type alleles with alleles containing random mutations, transformed cells were plated on SC-Ura and incubated at 30℃ for 48 hr. To confirm the success of each mutagenesis and to estimate actual mutation rates, we then sequenced the *RAP1* genes in 27 random colonies from the *RAP1* mutagenesis and we sequenced the second exon of *GCR1* in 18 random colonies from the *GCR1* mutagenesis. Next, 500 colonies from *RAP1* mutagenesis and 300 colonies from *GCR1* mutagenesis were streaked onto SC-Ura plates. After growth, one colony from each streak was patched on YPG and grown four days at 30℃. Then, patches were replica-plated with velvets onto SC + 5-FOA to eliminate sgRNA/Cas9 plasmids. Finally, 488 clones from *RAP1* mutagenesis and 355 clones from *GCR1* mutagenesis were arrayed in 96-well plates containing 0.5 ml of YPD (same plate design as used for the flow cytometry assays) and grown overnight at 30℃. 0.2 ml of cell culture from each well was then mixed with 46 µl of 80% glycerol in 96-well plates and stored at −80℃. The fluorescence of these strains was quantified by flow cytometry as described above to assess the impact of *RAP1* and *GCR1* mutations on $P_{TDH3}$-YFP expression (expression data for each mutant can be found in *Supplementary file 16*).

In our mutagenesis approach, we introduced a mutation that impaired the target PAM sequence in all *RAP1* and *GCR1* mutants. To determine the effect of this mutation alone, we generated strains YPW2701 and YPW2732 that carried the PAM mutation in the *RAP1* terminator or in the *GCR1* intron, respectively, without any other mutation in *RAP1* or *GCR1*. The fluorescence level of these two strains was not significantly different from the fluorescence level of the progenitor strain YPW1139 in flow cytometry assays.

## Estimation of *RAP1* and *GCR1* mutation rates

The expected number of mutations per PCR amplicon ($N_{mut}$) depends on the error rate of the Taq polymerase (μ), on the number of DNA duplications (*D*) and on the length of the amplicon (*L*): $N_{mut} = \mu \cdot D \cdot L$. The published error rate for a classic polymerase similar to DreamTaq is ~3 x 10$^{-5}$ errors per nucleotide per duplication (*McInerney et al., 2014*). Amplicon length was 3057 bp for *RAP1* mutagenesis and 2520 pb for *GCR1* mutagenesis. The number of duplications of PCR templates was calculated from the amounts of double stranded DNA quantified using Qubit 2.0 dsDNA assays before (*I*) and after (*O*) each mutagenic PCR reaction as follows: $D = ln\left(\frac{O}{I}\right) \div ln2$. For the first round of *RAP1* mutagenesis, $D = ln\left(\frac{6550}{1.93}\right) \div ln2 = 11.7$. For the second round of *RAP1* mutagenesis, $D = ln\left(\frac{3000}{68.1}\right) \div ln2 = 5.5$. Therefore, the total number of duplications was 17.2 and the expected number of mutations per amplicon $N_{mut}$ was 1.6 on average. For the first round of *GCR1* mutagenesis, $D = ln\left(\frac{6870}{2.15}\right) \div ln2 = 11.6$. For the second round of *GCR1* mutagenesis, $D = ln\left(\frac{6535}{1.65}\right) \div ln2 = 12.0$. Therefore, the total number of duplications was 23.6 and the expected number of mutations per amplicon $N_{mut}$ was 1.8 on average.

## Effects of mutations in purine biosynthesis genes on expression from different promoters

We compared the individual effects of three mutations in the purine biosynthesis pathway (*ADE2-C1477a*, *ADE5-G1715a* and *ADE6-G3327a*) on YFP expression driven by four different yeast promoters ($P_{TDH3}$, $P_{RNR1}$, $P_{STM1}$, and $P_{GPD1}$). Each mutation was introduced individually in the genomes of four parental strains described in *Hodgins-Davis et al., 2019* carrying either $P_{TDH3}$-*YFP* (YPW1139), $P_{RNR1}$-*YFP* (YPW3758), $P_{STM1}$-*YFP* (YPW3764), or $P_{GPD1}$-*YFP* (YPW3757) reporter gene at the *ho* locus. Site-directed mutagenesis was performed as described in the corresponding section (see above). The fluorescence of the four parental strains, of a non-fluorescent strain (YPW978) and of the 12 mutant strains (four reporter genes x three mutations) was quantified using a Sony MA-900 flow cytometer (the BD Accuri C6 instrument used for other fluorescence assays was not available due to Covid-19 shutdown) in three replicate experiments performed on different days. For each experiment, all strains were grown in parallel in culture tubes containing 5 ml of YPD and incubated at 30°C for 16 hr. Each sample was diluted to 1–2 x 10$^7$ cells/mL in PBS prior to measurement. At least 5 x 10$^4$ events were recorded for each sample using a 488 nm laser for YFP excitation and a 525/50 optical filter for the acquisition of fluorescence. At least 5 x 10$^4$ events were recorded for each sample. Flow cytometry data were then processed in R using functions from the *FlowCore* package and custom scripts available in *Source code 1*. After log-transformation of flow data, events considered to correspond to single cells were selected on the basis of their forward scatter height and width (FSC-H and FSC-W). Fluorescence values of single cells were then normalized to account for differences in cell size. Finally, the median fluorescence among cells was computed for each sample and averaged across replicates of each genotype.

## Statistical comparisons of *trans*-regulatory and nonregulatory mutations

We established a set of 69 *trans*-regulatory mutations that included 52 mutations with a p-value below 0.01 in the *G*-tests comparing the frequencies of mutant and reference alleles in low and high fluorescence bulks (see above) as well as 17 mutations identified by Sanger sequencing in the coding sequence of purine biosynthesis genes. In parallel, we established a set of 1766 nonregulatory mutations regarding $P_{TDH3}$-*YFP* expression that included mutations with a p-value above 0.01 in the *G*-tests comparing the frequencies of mutant and reference alleles in low and high fluorescence bulks (see above) and mutations that did not affect $P_{TDH3}$-*YFP* expression in single-site mutants. We performed statistical analysis to compare properties of *trans*-regulatory and nonregulatory mutations using RStudio v1.2.5019 (R scripts are in *Source code 2*). We used *G*-tests (*likelihood.ratio* function in *Deducer* package) to compare the following properties between *trans*-regulatory and nonregulatory mutations: (i) the frequency of G:C to A:T transitions, (ii) the frequency of indels, (iii) the frequency of aneuploidies, (iv) the distribution of mutations among chromosomes, (v) the frequency of mutations in coding, intronic and intergenic sequences, (vi) the frequency of synonymous, nonsynonymous and nonsense changes among coding mutations, (vii) the frequency of coding mutations in transcription factors, (viii) the frequency of coding mutations in the predicted *TDH3* regulatory

network (see below), (ix) the proportion of mutations in eQTL regions (see below). We used resampling tests to compare the frequencies of different amino acid changes caused by *trans*-regulatory and nonregulatory mutations in coding sequences. We computed for each possible amino acid change the observed absolute difference between (i) the proportion of coding *trans*-regulatory mutations causing the amino acid change and (ii) the proportion of nonregulatory mutations causing the amino acid change. Then, we computed similar absolute differences for 10,000 randomly permuted sets of *trans*-regulatory and nonregulatory mutations. The p-value for each amino acid change was calculated as the proportion of resampled absolute differences greater or equal to the observed absolute difference.

### *TDH3* regulatory network

The network of potential *TDH3* regulators shown on *Figure 4* was established using data available in July 2019 on the YEASTRACT (http://www.yeastract.com/) repository of regulatory associations between transcription factors and target genes in *Saccharomyces cerevisiae* (*Teixeira et al., 2018*). We used the tool 'Regulation Matrix' to obtain three matrices in which rows corresponded to the 220 transcription factor genes in YEASTRACT and columns corresponded to the 6886 yeast target genes included in the database. In the first matrix obtained using the option 'Only DNA binding evidence', an element had a value of 1 if the transcription factor at the corresponding row was reported in the literature to bind to the promoter of the target gene at the corresponding column and a value of 0 otherwise. The two other matrices were obtained using the option 'Only Expression evidence' with either 'TF acting as activator' or 'TF acting as inhibitor'. An element had a value of 1 only in the 'TF acting as activator' matrix if perturbation of the transcription factor at the corresponding row was reported to increase expression of the target gene at the corresponding column. An element had a value of 1 only in the 'TF acting as inhibitor' matrix if perturbation of the transcription factor at the corresponding row was reported to decrease expression of the target gene at the corresponding column. An element had a value of 1 in both matrices if perturbation of the transcription factor at the corresponding row was reported to affect expression of the target gene at the corresponding column in an undetermined direction. Finally, an element had a value of 0 in both matrices if perturbation of the transcription factor at the corresponding row was not reported to alter expression of the target gene at the corresponding column in the literature. We then used a custom R script (*Source code 2*) to generate a smaller matrix that only contained first level and second level regulators of *TDH3* and *TDH3* itself. A transcription factor was considered to be a first level regulator of *TDH3* if a regulatory association with *TDH3* was supported both by DNA binding evidence and expression evidence. A transcription factor was considered to be a second level regulator of *TDH3* if a regulatory association with a first level regulator of *TDH3* was supported both by DNA binding evidence and expression evidence. The network shown on *Figure 4* was drawn using Adobe Illustrator based on regulatory interactions included in the matrix of *TDH3* regulators (in *Supplementary file 12*). To determine whether mutations in the *TDH3* regulatory network constituted a significant mutational source of regulatory variation affecting $P_{TDH3}$ activity, we compared the proportions of *trans*-regulatory and non-regulatory mutations that were located in a *TDH3* regulator gene (first or second level) using a *G*-test (*likelihood.ratio* function in R package *Deducer*).

### Competitive fitness assays

We performed competitive growth assays to quantify the fitness of 62 strains with random mutations in the second exon of *GCR1*. These 62 strains corresponded to all *GCR1* mutants that showed a significant decrease of $P_{TDH3}$-YFP expression as quantified by flow cytometry as well as *GCR1* mutants for which *GCR1* exon 2 was sequenced and the location of mutations was known. The 62 strains were thawed on YPG plates as well as reference strains YPW1139 and YPW2732 and strain YPW1182 that expressed a GFP (Green Fluorescent Protein) reporter instead of YFP. After 3 days of incubation at 30°C, strains were arrayed in four replicate 96-well plates containing 0.5 ml of YPG per well. In parallel, the [GFP+] strain YPW1182 was also arrayed in four replicate 96-well plates. The eight plates were incubated on a wheel at 30°C for 32 hours. We then measured the optical density at 620 nm of all samples using a Sunrise plate reader (Tecan) and calculated the average cell density for each plate. Samples were then transferred to 1.2 ml of YPD in 96-well plates to reach an average cell density of $10^6$ cells/ml for each plate. 21.25 µl of samples from plates containing [YFP+] strains

were mixed with 3.75 µl of [GFP+] samples in four 96-well plates containing 0.45 ml of YPD per well. The reason why [YFP+] and [GFP+] strains were mixed to a 17:3 ratio is because we anticipated that some of the *GCR1* mutants may grow slower than the [GFP+] competitor in YPD. Samples were then grown on a wheel at 30°C for 10 hr and the optical density was measured again after growth to estimate the average number of generations for each plate. The ratio of [YFP+] and [GFP+] cells in each sample was quantified by flow cytometry before and after the 10 hr of growth. Samples were analyzed on a BD Accuri C6 flow cytometer with a 488 nm laser used for excitation and two different optical filters (510/10 and 585/40) used to separate YFP and GFP signals. FCS data were analyzed with custom R scripts using *flowCore* and *flowClust* packages (*Source code 1*) as described in *Duveau et al., 2018*. First, we filtered out artifactual events with extreme values of forward scatter or fluorescence intensity. Then, for each sample we identified two clusters of events corresponding to [YFP+] and [GFP+] cells using a principal component analysis on the logarithms of FL1.H and FL2.H (height of the fluorescence signal captured through the 510/10 and 585/40 filters, respectively). Indeed, [YFP+] cells tend to have lower FL1.H value and higher FL2.H value than [GFP+] cells and these two parameters are positively correlated. The competitive fitness of [YFP+] cells relative to [GFP+] cells was calculated as the exponential of the slope of the linear regression of $log_e\left(\frac{YFP}{GFP}\right)$ on the number of generations of growth (where *YFP* corresponds to the number of [YFP+] cells and *GFP* corresponds to the number of [GFP+] cells). We then divided the fitness of each sample by the mean fitness among all replicates of the reference strain YPW1139 to obtain a fitness value relative to YPW1139. The fitness of each strain was calculated as the mean relative fitness among the four replicate populations for that strain. These fitness data can be found in *Supplementary file 16*.

## Gene ontology (GO) analysis

GO term analyses were performed on http://www.pantherdb.org/ website in June 2020 (*Mi et al., 2019*). In 'Gene List Analysis', we used 'Statistical overrepresentation test' on a query list corresponding to the 42 genes affected by *trans*-regulatory coding mutations. GO enrichment was determined based on a reference list of the 1251 genes affected by non-regulatory coding mutations using Fisher's exact tests. Four separate analyses were performed for GO biological processes, GO molecular functions, GO cellular components and PANTHER pathways. GO terms that are significantly enriched in the list of *trans*-regulatory mutations (mutations associated with fluorescence level) relative to non-regulatory mutations (mutations not associated with fluorescence level) at $p < 0.05$ are listed in *Supplementary file 9*.

## Enrichment of mutations in eQTL regions

Genomic regions containing expression quantitative trait loci (eQTL) associated with $P_{TDH3}$-*YFP* expression variation in three different crosses (BYxYPS1000, BYxSK1 and BYxM22) were obtained from *Supplementary file 11* in *Metzger and Wittkopp, 2019*. A custom R script was used to determine the number of *trans*-regulatory and non-regulatory mutations located inside and outside these eQTL intervals (*Source code 2*). *G*-tests were performed to determine whether the proportion of *trans*-regulatory mutations in eQTL intervals was statistically different from the proportion of non-regulatory mutations in the same eQTL intervals.

## Data archiving

De-multiplexed sequencing data are available in FASTQ format from NCBI Sequence Read Archive (https://www.ncbi.nlm.nih.gov/sra) under BioProject number PRJNA706682. Flow cytometry data (FCS files) are available on the Flow Repository (https://flowrepository.org/) under the following experiments ID: FR-FCM-Z3WV for the secondary screen of fluorescence in EMS mutants shown in *Figure 1E*, FR-FCM-Z3JY for the quantifications of fluorescence in single site mutants and in the corresponding EMS mutants (*Figure 2E–G*), FR-FCM-Z3J2 for the quantifications of fluorescence in *RAP1* mutant strains (*Figure 5E*), FR-FCM-Z3J3 for the quantifications of fluorescence in *GCR1* mutant strains (*Figure 5F–G*) and FR-FCM-Z3J5 for the quantifications of fitness in the same *GCR1* mutants strains (*Figure 5G*).

## Acknowledgements

We thank Gaël Yvert and Mark Hill for helpful comments on the manuscript, the University of Michigan sequencing core and University of Michigan flow cytometry core for research support, and the National Institutes of Health (R01GM108826 and R35GM118073 to PJW), European Molecular Biology Organization (EMBO ALTF 1114–2012 to FD), National Science Foundation (MCB-1929737 to PJW), NIH Genetics Training grant (T32GM007544 to PVZ), NIH Genome Sciences Training Grant (T32HG000040 to BPHM and MAS), and the Michigan Life Sciences Fellow program (MAS) for funding.

## Additional information

### Competing interests

Patricia J Wittkopp: Senior editor, *eLife*. The other authors declare that no competing interests exist.

### Funding

| Funder | Grant reference number | Author |
| --- | --- | --- |
| National Institutes of Health | R01GM108826 | Patricia J Wittkopp |
| National Institutes of Health | R35GM118073 | Patricia J Wittkopp |
| European Molecular Biology Organization | 1114-2012 | Fabien Duveau |
| National Science Foundation | MCB-1929737 | Patricia J Wittkopp |
| National Institutes of Health | T32GM007544 | Petra Vande Zande |
| National Institutes of Health | T32HG000040 | Brian PH Metzger |
| National Institutes of Health | T32HG000040 | Mohammad A Siddiq |
| University of Michigan | Michigan Life Sciences Fellow program | Mohammad A Siddiq |

The funders had no role in study design, data collection and interpretation, or the decision to submit the work for publication.

### Author contributions

Fabien Duveau, Conceptualization, Data curation, Formal analysis, Supervision, Validation, Investigation, Visualization, Methodology, Writing - original draft, Writing - review and editing; Petra Vande Zande, Formal analysis, Validation, Investigation, Methodology, Writing - review and editing; Brian PH Metzger, Conceptualization, Investigation, Methodology, Writing - review and editing; Crisandra J Diaz, Elizabeth A Walker, Stephen Tryban, Bing Yang, Validation, Investigation; Mohammad A Siddiq, Formal analysis, Validation, Investigation, Methodology; Patricia J Wittkopp, Conceptualization, Supervision, Funding acquisition, Methodology, Writing - original draft, Project administration, Writing - review and editing

### Author ORCIDs

Fabien Duveau https://orcid.org/0000-0003-4784-0640
Brian PH Metzger http://orcid.org/0000-0003-4878-2913
Patricia J Wittkopp https://orcid.org/0000-0001-7619-0048

### Decision letter and Author response

Decision letter https://doi.org/10.7554/eLife.67806.sa1
Author response https://doi.org/10.7554/eLife.67806.sa2

# Additional files

## Supplementary files

• Source code 1. R scripts used for the analysis of flow cytometry data.

• Source code 2. R scripts used for the analysis of BSA-Seq data and for comparing the properties of *trans*-regulatory and non-regulatory mutations.

• Source code 3. R script used to annotate variants identified in BSA-Seq data.

• Source code 4. PBS script used to process FASTQ files.

• Source data 1. Compressed folder including 34. Source Data files in txt format that contain quantitative data displayed on *Figure 1B–E*, *Figure 2E–G*, *Figure 3A,B,E*, *Figure 5C–G*, *Figure 6*, *Figure 7*, *Figure 2—figure supplements 1–5*, *Figure 3—figure supplements 1–4* and *Figure 7—figure supplement 1*.

• Supplementary file 1. Sequencing depth in BSA-seq data.

• Supplementary file 2. List of all mutations identified by BSA-Seq or Sanger sequencing in this study.

• Supplementary file 3. Statistical associations between aneuploidies and fluorescence level.

• Supplementary file 4. Linked mutations associated with fluorescence level in BSA-Seq experiments.

• Supplementary file 5. Mutations identified by Sanger sequencing of candidate genes.

• Supplementary file 6. Mutations tested in single-site mutants.

• Supplementary file 7. Mutations associated with fluorescence level in BSA-Seq experiments.

• Supplementary file 8. Targeted mutagenesis of RAP1 residues making direct contact with DNA.

• Supplementary file 9. List of GO terms overrepresented in genes hit by causative mutations relative to genes hit by neutral mutations.

• Supplementary file 10. Mutations located in the coding sequence of glucose signaling genes.

• Supplementary file 11. *Trans*-regulatory effects of mutations in purine biosynthesis genes or iron homeostasis genes.

• Supplementary file 12. Files used as inputs for analyses performed with the PBS script (*Source code 4*) and R scripts (*Source code 1–3*).

• Supplementary file 13. List of DNA libraries grouped by sequencing runs.

• Supplementary file 14. List of oligonucleotides used in this study.

• Supplementary file 15. Construction of single-site mutant strains.

• Supplementary file 16. Phenotypes of RAP1 mutants (expression) and GCR1 mutants (expression and fitness).

• Transparent reporting form

## Data availability

Sequencing data have been deposited in NCBI SRA under BioProject code PRJNA706682. Flow cytometry data have been deposited in the FlowRepository (https://flowrepository.org/) under experiments ID FR-FCM-Z3WV, FR-FCM-Z3JY, FR-FCM-Z3J2, FR-FCM-Z3J3 and FR-FCM-Z3J5. All data generated or analysed during this study are included in Supplementary Files. Source data have been provided in Supplementary File 20 for Figure 1B-E, Figure 2E-G, Figure 3A,B,E, Figure 5C-G, Figure 6, Figure 7, Figure 2 - figure supplements 1-5, Figure 3 - figure supplements 1-2.

The following datasets were generated:

| Author(s) | Year | Dataset title | Dataset URL | Database and Identifier |
|---|---|---|---|---|
| Duveau F, Wittkopp PJ | 2021 | Mapping yeast trans-regulatory mutations by sequencing bulk segregant populations | https://www.ncbi.nlm. nih.gov/bioproject/ PRJNA706682 | NCBI BioProject, PRJNA706682 |
| Duveau F, Wittkopp | 2021 | Secondary screen of pTDH3-YFP | http://flowrepository. | FlowRepository, FR- |

| | | | | |
|---|---|---|---|---|
| PJ | | expression changes in yeast EMS mutants | org/id/FR-FCM-Z3WV | FCM-Z3WV |
| Duveau F, Wittkopp PJ | 2021 | Expression of pTDH3-YFP in trans-regulatory mutant strains of yeast | http://flowrepository. org/id/FR-FCM-Z3JY | FlowRepository, FR-FCM-Z3JY |
| Duveau F, Wittkopp PJ | 2021 | Expression of pTDH3-YFP in RAP1 mutant strains of yeast | http://flowrepository. org/id/FR-FCM-Z3J2 | FlowRepository, FR-FCM-Z3J2 |
| Duveau F, Wittkopp PJ | 2021 | Expression of pTDH3-YFP in GCR1 mutant strains of yeast | http://flowrepository. org/id/FR-FCM-Z3J3 | FlowRepository, FR-FCM-Z3J3 |
| Duveau F, Wittkopp PJ | 2021 | Fitness of GCR1 mutant strains of yeast in rich medium | http://flowrepository. org/id/FR-FCM-Z3J5 | FlowRepository, FR-FCM-Z3J5 |

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
