## [Decision Letter]

**Acceptance summary:**

The relationship between traits and mutations influences the rate and direction in which traits evolve. One key question in evolutionary biology is therefore how traits can be affected by spontaneous mutations. Here, the authors map a set of mutations that affect the expression of a focal gene in yeast, and examine their individual effects and location in the genome and in the regulatory network. The work is rigorous and the results are well presented. The findings will be of great interest for geneticist and evolutionary biologists interested in the evolution of gene expression and of complex traits.

**Decision letter after peer review:**

Thank you for submitting your article "Mutational sources of trans-regulatory variation affecting gene expression in *Saccharomyces cerevisiae*" for consideration by *eLife*. Your article has been reviewed by 3 peer reviewers, and the evaluation has been overseen by a Reviewing Editor and Molly Przeworski as the Senior Editor. The following individuals involved in review of your submission have agreed to reveal their identity: Fei He (Reviewer #2); Jukka-Pekka Verta (Reviewer #3).

Essential Revisions:

The three reviewers and I have appreciated the manuscript in terms of the questions addressed and of the quality of the work. However, we have identified several issues that would need to be addressed. The main points are:

1) One reviewer comments on the fact that the overlap with previous eQTLs is modest and not completely persuasive, for instance because both approaches could have similar biases and thus produce an enrichment. It would be important to clarify this point. Also, other comments relate to the fact that the overlap with known regulators and previous eQTLs was expected. It would be useful in the introduction to layout the reasons why natural variation (analyzed by eQTL mapping for instance) and more classical functional genetics studies, performed with gene deletions and mutants, would or would not overlap with spontaneous mutants such as the ones analyzed here. For instance, non-sense variants acting in trans can be recovered in experiments like yours but could be rare in nature because of selection acting directly on the genes or on pleiotropic effects. Many trans-regulatory relationships could therefore be missed. Such a justification (based on technical, biological and evolutionary considerations) regarding the need for this type of experiment would also broaden the scope of the paper.

2) One reviewer mentions that the inability to map some of the causal mutations may be due to the complex and polygenic architecture of such traits. It would be important to address this question in more details. The likelihood of detection of some types of variation may have a strong influence on the conclusions reached so it is important to eliminate or at least take into account these biases. The issue of potential interactions among mutations is also raised by the other reviewers.

3) As pointed out by reviewer 1, some of the result sections are very descriptive (sections on the number of mutations, their effects, etc.) and sometimes difficult to follow. A more summarized section with visual support could help.

4) Reviewer 2 raises potentially important questions about the statistical analyses and GO analysis. It would be important to answer those and make the appropriate changes when needed.

5) Reviewer 3 (and indirectly reviewers 1 and 2) raises questions regarding the generality of the findings for other organisms with different layers of regulation of gene expression. It would be important to address this point for the board readership of *eLife*.

*Reviewer #1 (Recommendations for the authors):*

I recommend greater consideration of the effect-size distribution of trans-acting mutations and acknowledgment that the mutations discovered here- mostly missense and nonsense mutations – may not be typical of the underlying spectrum of trait-relevant mutations. I thought in particular as I was reading of the very recent *eLife* paper from Sinnott-Armstrong et al., showing that molecular traits in humans are hugely polygenic, and that the individually-detectable genes are the ones you'd expect. I don't believe that discussing this viewpoint diminishes the authors' findings at all. It simply requires changing claims like coding mutations are "more likely to affect" to "more likely to be detected" or something like that.

The mutations discovered by candidate-gene resequencing obviously have a totally different ascertainment model than the ones discovered by mapping, and all analyses should be repeated without those (I think most are already).

I got quite lost in trying to track the different numbers of strains and mutations through the manuscript. Some kind of table or flowchart would really help. With some effort, I get that there are 82 mutants and 46 of these have BSA hits. Of these 46, 29 have one hit and so we can carry these 29 forward. 13 have two hits, and of these 5 strains give 2 unlinked hits, so that brings 10 more mutations forward. Of the remaining 12 strains with multiple linked mutations, 9 have different G statistics so that brings 9 forward. Now we're at 29+10+9 = 48. It's unclear how we end up with 12 linked-case mutations at line 293. I guess it's nine where there were differences in G statistics and 3 from the functional tests? The source of these last three is not explained very clearly. That gives me 51 mapped mutations (vs 52 at line 291) + 17 candidate gene mutations; I'm not sure how that turns into 66 total at line 291. Without belaboring the point, it's hard to keep track here. Similarly, I don't know where the 1766 non-trans mutations come from. There are 1819 total mutations and 69 causal mutations, leaving 1750 rather than 1766. And finally, I am confused by the report of 23.9 mutations per strain, which does not match 1819 mutations / 82 strains = 22.18.

*Reviewer #2 (Recommendations for the authors):*

In summary, this work is interesting and there is some weakness. It should be done with major revision as my opinion. As always there are a number of unclarities in the manuscript, I would like the authors to elucidate.

1) As the trans regulation could be polygenic selection, and this should be discussed in the paper, particularly there are 36 mutations with the small effects.

2) The authors tried to correlate the trans-regulatory mutations are enriched natural variation affecting TDH3 expression. However, the proof of finding is not so strong.

3) As the authors did a lot of test for the significance, I only found few cases of p value adjustment of post-hoc test.

4) RNAseq of realtime-PCR can test the expression change of trans regulators because of mutations. if this data is supplied, it will reveal trans changes of function or abundance effects on expression of targets.

5) The logical structure of the main text can be improved a bit.

6) In Figure 1B-C, the distribution of relative fluorescence level seems unequal distribution. Is it normal or biased?

7) In Figure 6B, how and why the authors selected these GO terms for presentation from 156 enriched GO terms in suppl. Data 8? Did the author do cluster for the GO terms? The GO of metabolism almost means everything, and I think it is not meaningful for biological questions.

8) I am not sure whether aneuploid can classify into trans regulation.

*Reviewer #3 (Recommendations for the authors):*

Line 34. Please be more specific about the role of trans-variation in "with expression variation often derived from trans-regulatory mutations within species."

Line 44. Please expand the thought in this last sentence to include implications, instead of simply stating that overlap exists.

Line 68. There exists studies analysing the mutational target size of trans-effects, it would be appropriate to cite them here.

Lines 134-136, 138. Please clarify the selection criteria used here – selected at random or are these all the available mutants (based on what criteria)?

Lines 290-295. By this point, you have referred to multiple sets of SNPs in the different analyses (chosen at random/by targeted analyses, showing association in BSA or not, tested for functional effects or not etc.) and at least I could not follow how the analyses resulted in the summarizing numbers of SNPs presented in this paragraph. I don't have a perfect suggestion to address this problem, but I think a schematic figure would go a long way keep the reader on track.

Lines 290-291. Is it possible to put these figures into perspective of the whole mutagenesis experiment? What is the total number of mutations created in the experiment, how many of these influence Ptdh3-YFP and how many further have trans effects?

Line 297. What's the criteria to select exactly these 1766 mutations?

Lines 685-698. The details explaining the choice of the statistical test would better fit a supplementary note than the methods.

Lines 515-546 (the whole section). Can you observe any segregating variation in RAP1 or GCR1? An absence of natural genetic variation could indicate that trans effects cannot happen in these genes because of their lethality (case of RAP1), while observable variation could strengthen the case that absence of trans effects is the result of sampling bias (case of GCR1).

Figure legends (esp. Figure 2 and its supplements). The figure legends are very long and contain e.g. description of methods and results. I would consider condensing the legends as much as possible or alternatively explaining the additional analyses in the supplementary materials. On the other hand I do understand the logic of explaining e.g. supporting analyses in figure supplements and their legends, so I'll leave the choice to the editor and the authors.

Figure 4. You conclude that "it seems that regulatory networks describing the relationships between transcription factors and target genes might capture only a small fraction of the potential sources of trans-regulatory variation." – yet, figure 4 describes exactly these kinds of relationships between transcription factors and downstream genes. Would it be possible to illustrate the significance of other trans-acting sources in addition to transcription factors in figure 4?

---

## [Author Response]

Essential Revisions:The three reviewers and I have appreciated the manuscript in terms of the questions addressed and of the quality of the work. However, we have identified several issues that would need to be addressed. The main points are:1) One reviewer comments on the fact that the overlap with previous eQTLs is modest and not completely persuasive, for instance because both approaches could have similar biases and thus produce an enrichment. It would be important to clarify this point.

We respond to this issue more fully below in response to reviewer 1’s more detailed

comment. In short, we tested whether differences in sequencing depth across the genome, which might be similar in both BSA-seq mapping experiments, could explain the reported overlap between trans-regulatory mutations and eQTLs. We found that variation in sequencing depth was unlikely to explain the observed enrichment of trans-regulatory mutations in regions of the genome previously shown to harbor eQTLs affecting TDH3 expression. A new figure (Figure 7 – supplementary figure 1) has been added to the manuscript showing the results of this analysis.

Also, other comments relate to the fact that the overlap with known regulators and previous eQTLs was expected.

We agree that some overlap was expected (which is why we tested for the enrichment);

however, the degree to which the trans-regulatory mutants overlapped with known regulators and previous eQTL was not predictable and, to the best of our knowledge, has not previously been tested empirically. For example, we found that only 6% of trans-regulatory mutations mapped to transcription factors previously shown to regulate TDH3; a priori, we would have guessed that this overlap would have been much greater. In fact, we failed to recover mutations in the two best characterized direct regulators of TDH3 (Rap1p and Gcr1p), which we further investigated in the paper. As stated in the paper, this observation suggests that:

“regulatory networks describing the relationships between transcription factors and target genes might capture only a small fraction of the potential sources of transregulatory variation.”

Nonetheless, we have tried to better convey the expected overlap between known regulators and the trans-regulatory mutants by modifying the sentence on line 448 to read:

“Therefore, the inferred regulatory network had predictive power as expected, but the vast majority of trans-regulatory coding mutations (61 of 65, or 94%) mapped to genes outside of this network.”

As for the overlap with eQTL, we agree that an exhaustive study of mutations affecting

expression of TDH3 should overlap with eQTL affecting TDH3 expression, but it was less clear that the set of 69 mutations we mapped would be sufficient to see an enrichment. For example, if the mutations we mapped were more likely to be deleterious than the variants responsible for the eQTL mapped between strains (for instance if they tended to have larger or more pleiotropic effects), we might not have seen a significant overlap. We might also not have seen an overlap if epistatic interactions among natural variants were required to impact expression of TDH3 expression because we only looked at the effects of single trans-regulatory mutations in this study. These points have been added to the revised Discussion (which was a Conclusions section in the original submission), as suggested in the rest of the editor’s comment shown below.

It would be useful in the introduction to layout the reasons why natural variation (analyzed by eQTL mapping for instance) and more classical functional genetics studies, performed with gene deletions and mutants, would or would not overlap with spontaneous mutants such as the ones analyzed here. For instance, non-sense variants acting in trans can be recovered in experiments like yours but could be rare in nature because of selection acting directly on the genes or on pleiotropic effects. Many trans-regulatory relationships could therefore be missed. Such a justification (based on technical, biological and evolutionary considerations) regarding the need for this type of experiment would also broaden the scope of the paper.

As requested, we have now added such a discussion to the introduction in a new second

paragraph (lines 68-80) and by modifying the end of the third paragraph and the beginning of the fourth paragraph (lines 93-105). We have also addressed these topics in the revised discussion (lines 640-662).

2) One reviewer mentions that the inability to map some of the causal mutations may be due to the complex and polygenic architecture of such traits. It would be important to address this question in more details.

While we agree that natural variation in gene expression has a complex and polygenic

architecture, we mapped mutations in this study from mutant genotypes with only ~24

mutations spread throughout the entire genome (compared to ~33,000 to ~56,000 SNPs, or ~2.8 to 4.6 SNP/kb, between strains of *S. cerevisiae* used in the eQTL mapping study). That is, we intentionally designed this experiment to try to capture changes in activity of the TDH3 promoter caused by single mutations. Nonetheless, our inability to map some of the causal mutations might have been due to their small individual effects or interactions between mutations, which are possibilities described and explored in paragraph on lines 235-249 and in Figure 2 —figure supplement 2 and 3. We also added the following sentence on line 149 to try to make this aspect of the experimental design more clear:

“The dose of EMS used in these studies was chosen so that most mutants with a detectable change in PTDH3-YFP expression should have only one mutation causing this change in expression among the mutations they carry (Metzger et al., 2016; Gruber et al., 2012).”

The likelihood of detection of some types of variation may have a strong influence on the conclusions reached so it is important to eliminate or at least take into account these biases.

To address this concern, which we assume to be primarily related to the effect size of

mutations we could map as expanded upon below by reviewer 1, we have modified the text to limit the scope of our conclusions (e.g., line 383) and included this limitation of the work in the revised discussion (lines 666-669).

The issue of potential interactions among mutations is also raised by the other reviewers.

We have added text to the discussion acknowledging the importance of considering

epistasis when studying regulatory variation and identifying this as an area ripe for future study (lines 656-662). We think our data have little to say about genetic interactions because most mutant phenotypes we studied were caused by single mutations. Specifically, the individual mapped mutations explained 94% of the variation in expression observed among the original EMS mutants (Figure 2G), each of which carried an additional 20-30 other mutations. These data suggest that for the mutants we examined, epistatic interactions had negligible effects on expression of P_TDH3_-YFP.

3) As pointed out by reviewer 1, some of the result sections are very descriptive (sections on the number of mutations, their effects, etc.) and sometimes difficult to follow. A more summarized section with visual support could help.

We thank reviewers 1 and 3 for suggesting the idea of a flowchart to help tracking the

number of mutations and mutants at different steps of the study. We have followed their suggestion and added such a diagram as a new Figure 1 —figure supplement 1.

4) Reviewer 2 raises potentially important questions about the statistical analyses and GO analysis. It would be important to answer those and make the appropriate changes when needed.

We respond more fully to these points in response to reviewer 2’s comments below.

5) Reviewer 3 (and indirectly reviewers 1 and 2) raises questions regarding the generality of the findings for other organisms with different layers of regulation of gene expression. It would be important to address this point for the board readership of eLife.

To fully address this comment, we expanded the former Conclusions section into a more

complete Discussion section and specifically addressed issues related to generalizability in the final paragraph (lines 682-697).

Reviewer #1 (Recommendations for the authors):I recommend greater consideration of the effect-size distribution of trans-acting mutations and acknowledgment that the mutations discovered here- mostly missense and nonsense mutations – may not be typical of the underlying spectrum of trait-relevant mutations. I thought in particular as I was reading of the very recent eLife paper from Sinnott-Armstrong et al., showing that molecular traits in humans are hugely polygenic, and that the individually-detectable genes are the ones you'd expect. I don't believe that discussing this viewpoint diminishes the authors' findings at all. It simply requires changing claims like coding mutations are "more likely to affect" to "more likely to be detected" or something like that.

We thank the reviewer for this comment. We have now revised the introduction to make more explicit the differences between mutations identified and characterized in this study and variation segregating in the wild. We have also made changes to qualify some statements, such as that coding mutations are more likely to affect expression by 3% or more (line 383), which limits the scope of the statement to the size of effects we were able to examine. Finally, we have added explicit statements about the limitations of the study to the revised discussion.

The mutations discovered by candidate-gene resequencing obviously have a totally different ascertainment model than the ones discovered by mapping, and all analyses should be repeated without those (I think most are already).

We have now repeated all analyses without the 17 mutations identified by sequencing

candidate genes. These results are reported in Figure 3 —figure supplement 1, Figure 3 - figure supplement 2, Figure 3 —figure supplement 4, Supplementary File 8 and in multiple parts of the Results section. The same patterns were observed and the significance of statistical tests was not affected by the removal of the 17 mutations, thus our conclusions did not change. In particular, the results of the GO enrichment analysis were very similar because this analysis was focused on genes and not on mutations: only two genes were excluded (ADE2 and ADE6) when we removed the 17 mutations identified by sequencing candidate genes.

I got quite lost in trying to track the different numbers of strains and mutations through the manuscript. Some kind of table or flowchart would really help. With some effort, I get that there are 82 mutants and 46 of these have BSA hits. Of these 46, 29 have one hit and so we can carry these 29 forward. 13 have two hits, and of these 5 strains give 2 unlinked hits, so that brings 10 more mutations forward. Of the remaining 12 strains with multiple linked mutations, 9 have different G statistics so that brings 9 forward. Now we're at 29+10+9 = 48. It's unclear how we end up with 12 linked-case mutations at line 293. I guess it's nine where there were differences in G statistics and 3 from the functional tests? The source of these last three is not explained very clearly. That gives me 51 mapped mutations (vs 52 at line 291) + 17 candidate gene mutations; I'm not sure how that turns into 66 total at line 291. Without belaboring the point, it's hard to keep track here. Similarly, I don't know where the 1766 non-trans mutations come from. There are 1819 total mutations and 69 causal mutations, leaving 1750 rather than 1766. And finally, I am confused by the report of 23.9 mutations per strain, which does not match 1819 mutations / 82 strains = 22.18.

We appreciate the efforts made by the reviewer to point out numbers for which the sources needed to be clarified. These comments helped us design a diagram showing the number of mutations and mutants included at each step of the study (Figure 1 —figure supplement 1). We hope this diagram will help readers to track these numbers throughout the manuscript much more easily.

Reviewer #2 (Recommendations for the authors):In summary, this work is interesting and there is some weakness. It should be done with major revision as my opinion. As always there are a number of unclarities in the manuscript, I would like the authors to elucidate.1) As the trans regulation could be polygenic selection, and this should be discussed in the paper, particularly there are 36 mutations with the small effects.

We agree that trans-regulatory variation segregating in the wild is generally polygenic and have revised the introduction to include this point more specifically. Indeed, our group has previously shown that trans-regulatory variation affecting TDH3 promoter activity in *S. cerevisiae* is highly polygenic (Metzger et al., 2019 Evolution Letters). However, the goal for this work was to identify individual, trans-regulatory mutations affecting TDH3 promoter activity. For this reason, the genotypes used for mapping trans-regulatory mutations were generated with a low dose of EMS that introduced on average 24 mutations per genome, compared to ~34,000 genetic differences between the BY and M22 strains of *S. cerevisiae* and ~54,000 genetic differences between BY and either SK1 or YPS1000. In most cases, only one of these ~24 mutations fully explained the changes in TDH3 expression seen in the mutant. For the 36 mutants in which we were unable to identify a single causative mutation, small effect sizes are one of the possible explanations, as described more fully in the paragraph on lines 235-249.

2) The authors tried to correlate the trans-regulatory mutations are enriched natural variation affecting TDH3 expression. However, the proof of finding is not so strong.

As described above in response to reviewer 1, we have included additional analyses to test for possible biases associated with BSA-seq in the two studies that might contribute to the observed statistically significant overlap in the revised manuscript. We are also not sure what the reviewer means by “not so strong” in this context; the overlap reported was supported statistically by a p-value of P = 9.6 x 10-^5^.

3) As the authors did a lot of test for the significance, I only found few cases of p value adjustment of post-hoc test.

We are guessing that the reviewer is asking about multiple testing corrections rather than post-hoc tests, as we used a false discovery rate correction for multiple tests in Figure 2-supplement 5A. Although we did not use a multiple test correction for the BSA-seq data, we used a conservative significance threshold of 0.001 that was expected to result in a 3.5% false positive rate. Perhaps more importantly, we functionally validated the effects of 40 of the 41 associated mutations tested.

4) RNAseq of realtime-PCR can test the expression change of trans regulators because of mutations. if this data is supplied, it will reveal trans changes of function or abundance effects on expression of targets.

We agree that it would be interesting to know whether the mutations mapped also affected expression of the trans-regulators, but we do not think the additional experiments that would be required to address this point are necessary to support the conclusions presented.

5) The logical structure of the main text can be improved a bit.

We hope that the changes made to the revised manuscript, including a new flow chart added as Figure 1 —figure supplement 1, will help readers follow the work more easily.

6) In Figure 1B-C, the distribution of relative fluorescence level seems unequal distribution. Is it normal or biased?

Distributions of mutational effects for trans-regulatory mutations impacting gene expression have previously been described for the TDH3 promoter (Metzger et al., 2016) as well as for promoters from other genes (Hodgins-Davis et al., 2019). None of these distributions of mutational effects are consistent with a normal distribution. The distribution of mutational effects on P_TDH3_-YFP expression was found to have lower kurtosis (more mutations with large effects) than a normal distribution but was symmetrical (no significant skew, which is what we think the reviewer means by bias).

7) In Figure 6B, how and why the authors selected these GO terms for presentation from 156 enriched GO terms in suppl. Data 8? Did the author do cluster for the GO terms? The GO of metabolism almost means everything, and I think it is not meaningful for biological questions.

We did not represent all 156 enriched GO terms on Figure 6B both to make the figure easier to read and because many GO terms were redundant due to the hierarchical structure of GO terms, as described above in response to this reviewer’s public comments. Instead, we only represented the most specific GO terms corresponding to the end tips of the GO hierarchy. We then further grouped enriched GO terms into four larger categories that are not formal GO terms to provide a broader overview of the shared properties of trans-regulatory mutations (all statistical analyses were performed with the formal GO terms, not on the four broader categories).

We think metabolism is a meaningful category because (1) it has been widely defined in the literature, (2) it is a commonly used term, and (3) some, but not all, genes in this group encode proteins involved in metabolic pathways. We also think it makes a meaningful distinction between, for example, genes encoding enzymes acting in the de novo purine biosynthesis pathway (ADE2, ADE4, ADE5 and ADE6), which are clearly involved in metabolism, and genes encoding proteins involved in other biological processes such as nucleosome positioning (TUP1 and CHD1), which are not always involved in metabolism.

8) I am not sure whether aneuploid can classify into trans regulation.

A trans-regulatory mutation is defined as any mutation that impacts expression of both

alleles of a gene in diploid cells. Such effects are generally mediated by diffusible molecules such as RNAs or proteins. Aneuploidies are thus classified as trans-acting mutations when they impact expression of genes located on other chromosomes (as is the case in this study) because their impacts are presumably caused by changes in the abundance of trans-acting proteins and/or RNAs produced from genes on the aneuploid chromosome.

Reviewer #3 (Recommendations for the authors):Line 34. Please be more specific about the role of trans-variation in "with expression variation often derived from trans-regulatory mutations within species."

This part of the sentence has been removed in the revised abstract to avoid potential

confusion.

Line 44. Please expand the thought in this last sentence to include implications, instead of simply stating that overlap exists.

This sentence has been completely re-written in the revised abstract. We think that the

revised version addresses this comment as well as the previous comment while also helping to clarify the relationship between trans-regulatory mutations and trans-regulatory variation segregating within a species.

Line 68. There exists studies analysing the mutational target size of trans-effects, it would be appropriate to cite them here.

Empirical data measuring the target size for trans-regulatory mutations is very limited (e.g., Metzger et al., 2016); however, the expectation for a large mutational target size for transregulatory mutations has been described by many authors. This expectation is derived from the fact that each gene’s expression is influenced by proteins and RNAs encoded by many other genes and is described most fully in Hill et al., (2021). A citation to this review has been added to this sentence.

Lines 134-136, 138. Please clarify the selection criteria used here – selected at random or are these all the available mutants (based on what criteria)?

We added the following sentence on line 172 to clarify how the mutants were selected:

“Overall, the 82 mutants were selected randomly from the 528 EMS mutants that showed statistically significant fluorescence changes greater than 1% relative to wild-type (P < 0.05, see Methods and Figure 1 legend for a description of the statistical tests).”

Lines 290-295. By this point, you have referred to multiple sets of SNPs in the different analyses (chosen at random/by targeted analyses, showing association in BSA or not, tested for functional effects or not etc.) and at least I could not follow how the analyses resulted in the summarizing numbers of SNPs presented in this paragraph. I don't have a perfect suggestion to address this problem, but I think a schematic figure would go a long way keep the reader on track.

The revised version of the manuscript includes a new diagram (Figure 1 —figure supplement 1) that shows how many mutations and mutant strains were included at each step of the study.

Lines 290-291. Is it possible to put these figures into perspective of the whole mutagenesis experiment? What is the total number of mutations created in the experiment, how many of these influence Ptdh3-YFP and how many further have trans effects?

We hope that the new diagram Figure 1 —figure supplement 1 clarifies how the numbers mentioned in the original lines 290-291 were obtained from the mutagenesis experiments. The questions posed about the whole mutagenesis experiment are addressed in the Gruber et al. 2012 and Metzger et al., 2016 papers from which the mutants analyzed here were described. Both of these papers attempt to answer these questions, but the questions are harder to answer than they might seem and require making some assumptions about the data. Answering these questions definitively would require completing the sequencing and mapping described here for 82 mutants for all nearly 2000 EMS mutants analyzed in those studies.

Line 297. What's the criteria to select exactly these 1766 mutations?

We added the following sentences at the end of the paragraph on line 336 to clarify how the 1766 non-regulatory mutations were selected:

“To identify trends in the properties of these 69 trans-regulatory mutations, we compared them to 1766 mutations considered non-regulatory regarding PTDH3-YFP expression because they showed no significant association with expression of the reporter gene in the BSA-Seq experiment (G-test: P > 0.01, Figure 1 —figure supplement 1). To be more conservative, 8 mutations that showed a marginally significant association with expression (G-test: 0.001 < P < 0.01) as well as 15 mutations associated with expression only because of genetic linkage were excluded from further analyses.”

Lines 685-698. The details explaining the choice of the statistical test would better fit a supplementary note than the methods.

While we agree with the reviewer that these lines are not strictly methodological, we do think they are needed to justify the analysis methods used. We could certainly move them to a supplementary note, but we prefer to keep them here to prevent the reader from needing to flip back and forth between sections and hope that this choice is within the author’s discretion.

Lines 515-546 (the whole section). Can you observe any segregating variation in RAP1 or GCR1? An absence of natural genetic variation could indicate that trans effects cannot happen in these genes because of their lethality (case of RAP1), while observable variation could strengthen the case that absence of trans effects is the result of sampling bias (case of GCR1).

We could search for segregating variation in the coding sequences of RAP1 and GCR1

using genomic sequences for strains of *S. cerevisiae*, but without experimentally testing their effects it would not be possible to know whether any variants that exist affect activity of the TDH3 promoter. Our data show that several mutations in RAP1 and GCR1 are viable, so finding RAP1 and GCR1 variants segregating in natural populations would not be surprising or easy to interpret.

Figure legends (esp. Figure 2 and its supplements). The figure legends are very long and contain e.g. description of methods and results. I would consider condensing the legends as much as possible or alternatively explaining the additional analyses in the supplementary materials. On the other hand I do understand the logic of explaining e.g. supporting analyses in figure supplements and their legends, so I'll leave the choice to the editor and the authors.

We appreciate the reviewer pointing this out. We have edited the Figure 2 legend to shorten it and remove redundancy with the main text and methods. We also revisited the long Figure 2 —figure supplement 5 legend. In this case, we opted to leave it as is rather than create a separate supplementary text section that refers to this figure. We think that the current format will allow the reader to more easily follow the different hypotheses being tested with the different analyses. Because legends for figure supplements appear online only (i.e., not in the PDF version of the paper), we hope that the extra length there will be less problematic.

Figure 4. You conclude that "it seems that regulatory networks describing the relationships between transcription factors and target genes might capture only a small fraction of the potential sources of trans-regulatory variation." – yet, figure 4 describes exactly these kinds of relationships between transcription factors and downstream genes. Would it be possible to illustrate the significance of other trans-acting sources in addition to transcription factors in figure 4?

Figure 4 was specifically designed to test whether trans-regulatory mutations tended to lie in transcription factors previously implicated in regulating TDH3 expression, with edges in the network representing direct binding of a transcription factor to a target gene. Because the molecular mechanism by which most other genes harboring trans-regulatory mutations impact expression of TDH3 is not clear, we think it would be challenging to add these other types of genes into the figure without creating confusion. However, to better convey that this figure illustrates only a subset of the trans-regulatory mutations and non-regulatory mutations included in this study, we have modified the figure to show the number of total trans-regulatory and non-regulatory mutations included in the figure.